# Restructuring highly electron-deficient metal-metal oxides for boosting stability in acidic oxygen evolution reaction

Xinghui Liu [1,2], Shibo Xi [3], Hyunwoo Kim[4], Ashwani Kumar[1,2], Jinsun Lee[1,2], Jian Wang [5], Ngoc Quang Tran[1], Taehun Yang[1,2], Xiaodong Shao[1,2], Mengfang Liang[1,2], Min Gyu Kim [6] & Hyoyoung Lee [1,2,7 ✉]

The poor catalyst stability in acidic oxidation evolution reaction (OER) has been a long-time issue. Herein, we introduce electron-deficient metal on semiconducting metal oxides-consisting of Ir (Rh, Au, Ru)-MoO$_3$ embedded by graphitic carbon layers (IMO) using an electrospinning method. We systematically investigate IMO's structure, electron transfer behaviors, and OER catalytic performance by combining experimental and theoretical studies. Remarkably, IMO with an electron-deficient metal surface (Ir$^{x+}$; x > 4) exhibit a low overpotential of only ~156 mV at 10 mA cm$^{-2}$ and excellent durability in acidic media due to the high oxidation state of metal on MoO$_3$. Furthermore, the proton dissociation pathway is suggested via surface oxygen serving as proton acceptors. This study suggests high stability with high catalytic performance in these materials by creating electron-deficient surfaces and provides a general, unique strategy for guiding the design of other metal-semiconductor nanocatalysts.

[1] Center for Integrated Nanostructure Physics (CINAP), Institute of Basic Science (IBS), 2066 Seoburo, Jangan-Gu, Suwon 16419, Republic of Korea. [2] Department of Chemistry, Sungkyunkwan University (SKKU), 2066 Seoburo, Jangan-Gu, Suwon 16419, Republic of Korea. [3] Institute of Chemical and Engineering Sciences, A*STAR, 1 Pesek Road, Jurong Island 627833, Singapore. [4] Department of Energy Science, Sungkyunkwan University (SKKU), 2066 Seoburo, Jangan-Gu, Suwon 16419, Republic of Korea. [5] Department of Chemistry, College of Science, Seoul National University, Seoul 08826, Republic of Korea. [6] Beamline Research Division, Pohang Accelerator Laboratory (PAL), Pohang University of Science and Technology, Pohang 37673, Republic of Korea. [7] Department of Biophysics, Sungkyunkwan University (SKKU), 2066 Seoburo, Jangan-Gu, Suwon 16419, Republic of Korea. ✉email: hyoyoung@skku.edu

Hydrogen ($H_2$) fuel, as a clean energy carrier, is promising to provide an environmentally benign solution for global energy needs[1,2]. Among different ways of producing $H_2$, electrochemical water splitting plays a vital role in utilizing renewable energy sources[3–6]. Though the alkaline water electrolysis technology is dominating the large-scale production of $H_2$, proton exchange membrane (PEM) water electrolysis has clear advantages such as compact configuration, larger maximum current densities, higher energy efficiency, less $H_2$ impurity, and dynamic flexibility of operation[7–10]. As a half-reaction of water splitting, oxygen evolution reaction (OER) is a major bottleneck due to its sluggish kinetics, while the current OER catalysts typically degrade rapidly under acidic conditions, are not stable in highly oxidative environments and are of high cost[11–13]. Thus, developing low-cost and high-efficiency OER catalysts, especially those stable in acidic media, has been a pressing need but remains a grand challenge[14].

Significant progress has been achieved in developing active OER catalysts, though the stability under acidic conditions is still a big issue[15–18]. Among others, the first-row (3d) transition-metal oxides showed good promise as OER catalysts. For instance, Smith and et al. developed the amorphous metal oxide (a-$Fe_{100-y-z}Co_yNi_zO_x$) materials[17], and Friebel et al. investigated (Ni, Fe) oxyhydroxides layer structures[19]. Xin Wang and coworkers proposed a lattice oxygen oxidation mechanism pathway using metal oxyhydroxides, when two adjacent oxidized oxygen atoms can hybridize their oxygen holes without sacrificing metal-oxygen hybridization[20]. Recently, Sargent and coworkers suggested that modulating the 3d transition metal in metal (CoFe) oxyhydroxides by suitable transition metal (W) doping may provide further avenues to OER optimization[21]. However, only the performance of the state-of-the-art Ir-based and Ru-based catalysts have been expected to improve their catalytic activity and stability further[22–24]. Notably, Compared to Ru-based catalysts, Ir-based catalysts show higher stability and lower OER activity under acidic conditions[14,25–27]. Therefore, modulating the Ir-based catalysts to achieve enhanced OER activity, while simultaneously preserving high acid-stability serves are a promising route to develop OER catalysts suitable for large-scale applications.

The large-scale density functional theory (DFT) computations and emerging machine-learning techniques are greatly accelerating the innovation and discovery of catalysts[15,28,29]. Nørskov and coworkers identified 68 acid-stable candidates, such as Sb, and Mo, from 47, 814 nonbinary metal oxides for OER[16]. Ulissi and coworkers performed systematic high-throughput calculations to discover catalysts that could replace state-of-the-art iridium oxide catalysts. Besides, the theoretical finding that bound Mo-Ir oxides system has high acid-stability potential has not been experimentally reported in the literature[15]. In addition, some practical advantages demonstrated that the Ir with a high valence state is responsible for the high OER. For instance, using in situ and ex situ x-ray spectra, Juan-Jesús Velasco-Vélez et al. investigated the electrochemically active iridium nanoparticles for OER in acidic conditions and revealed that the catalytic activity is from the formation of shared electron-holes in the O 2p and Ir 5d, which leads to the generation of electron-deficient oxygen species[30]. Du and coworkers synthesized an $Ag_1/IrO_x$ single-atom catalyst, uncovering the high-valence $Ir^{x+}$ ($x > 4$) is responsible for the high catalytic OER performance[31]. Recently, Juan-Jesús Velasco-Vélez et al. probed clearly that the oxidation state of Ir during OER is >IV rather than that of Ir (III) because there is an increase in the intensity of the Ir-$L_3$ white-line peak due to the formation of more electron-holes in the Ir 5d orbitals[32].

To overcome the high stability with high activity, we report the highly electron-deficient metal on semiconducting metal oxides consisting of mixed Ir and $MoO_3$ embedded by graphitic carbon layers (IMO). The aim is to develop a highly active electrocatalyst with stability for OER in acidic conditions. To rationally design electron-deficient metal on metal oxides, the electrospinning strategy aid of polyvinylpyrrolidone (PVP) facilitates the different reduction ability of the two metal oxides because their formation energies are $-0.862$ eV/atom for $IrO_2$[33] and $-1.929$ eV/atom for $MoO_3$[15,34]. PVP was adopted to help reduce the $IrO_2$-only in the air annealing condition (500 °C) to achieve the electron-deficient surface of Ir, and to provide graphitic carbon layers from thermal decomposition. The graphitic carbon layer acted as the protective layer to confer high durability and conductivity to facilitate the fast electron transfer during OER process. IMO nanocomposite can therefore be successfully synthesized via the economical one-pot to create an electron-deficient surface on Ir ($Ir^{x+}$; $x > 4$) by virtue of two factors: (i) surface oxygen of Ir; (ii) the electron-withdrawing material of $MoO_3$. The IMO demonstrated superior OER activity by evidence of ultra-low overpotential and high stability compared to the benchmark materials of Ir and $RuO_2$, since the synergic effect of high surface state of Ir with the help of the $Mo^{5+}$ can withstand resistance in an oxidation state. This study not only uncovers the rational design of metal-metal oxides such as Ir (Rh, Au, Ru)-$MoO_3$ with superior catalytic performance by creating the electron-deficient surface, but also provides the general strategy-electron-deficient surface of metal on metal oxides driven by surface oxygen and electron-withdrawing groups of the substrate-for guiding other metal-semiconductor design.

## Results and discussion

### Principle and synthesis of metal-metal oxides—Ir (Rh, Au, Ru)-$MoO_3$—catalysts.
To fabricate the IMO with electron-deficient surface, the electrospinning synthetic scheme was designed for the IMO precursor (Fig. 1a). Consequently, the nanocomposite fiber of IMO precursor is formatted in the rotating collector using PVP substrate. PVP, firstly proposed as "polyol reduction" for synthesizing metallic Co and Ni from the oxide valence state by Figlarz et al.[35], has been assigned as a steric stabilizer or capping agent to synthesize various nanocrystals[36–38]. To the best of our knowledge, PVP was first applied to fabricate the metal (Ir, Rh, Ru, and Au)-semiconductor ($MoO_3$) nanocomposites (Fig. S1). The different formation energies for these metal-oxides ($-0.281$ eV/atom for $Au_2O_3$[39]; $-0.862$ eV/atom for $IrO_2$[33]; $-0.917$ eV/atom for $Rh_2O_3$[40]; $-1.202$ eV/atom for $RuO_2$[41]; $-1.929$ eV/atom for $MoO_3$[15][34] can form the basis for a general and unique strategy for designing metal-semiconductor catalysts. Note that the thermal decomposition of PVP also acting as graphitic carbon layers resulted in high conductivity, facilitating electron transfer during the OER process with high stability[42]. The nanorods of IMO, a representative metal-semiconductor, were successfully fabricated by means of a pot economy and one-pot synthesis, forming the three-dimensional (3D) network structures observed from the scanning electron microscope (SEM) and the annealing diagram (Fig. 1b). As expected, we got the nano-composited IMO in which $IrO_2$ is reduced to Ir, and $MoO_3$ is formed as the oxide state. The annealing procedure at 500 degrees in the air can induce the electron-deficient surface of metallic Ir due to the surface oxygen and heterointerface junction with the semiconductor of $MoO_3$[43]. More detailed evidence is given as follows.

### Characterization of morphologies and structures.
X-ray diffraction (XRD) patterns of IMO (Fig. 2a) indicated that phase properties of iridium are metallic as assumed above. The XRD pattern of $IrO_2$-$MoO_3$ (IOMO) indicated that the bi-metallic oxidized was synthesized successfully by comparison with IMO catalyst (Fig. S2). The morphology and distribution of IMO are

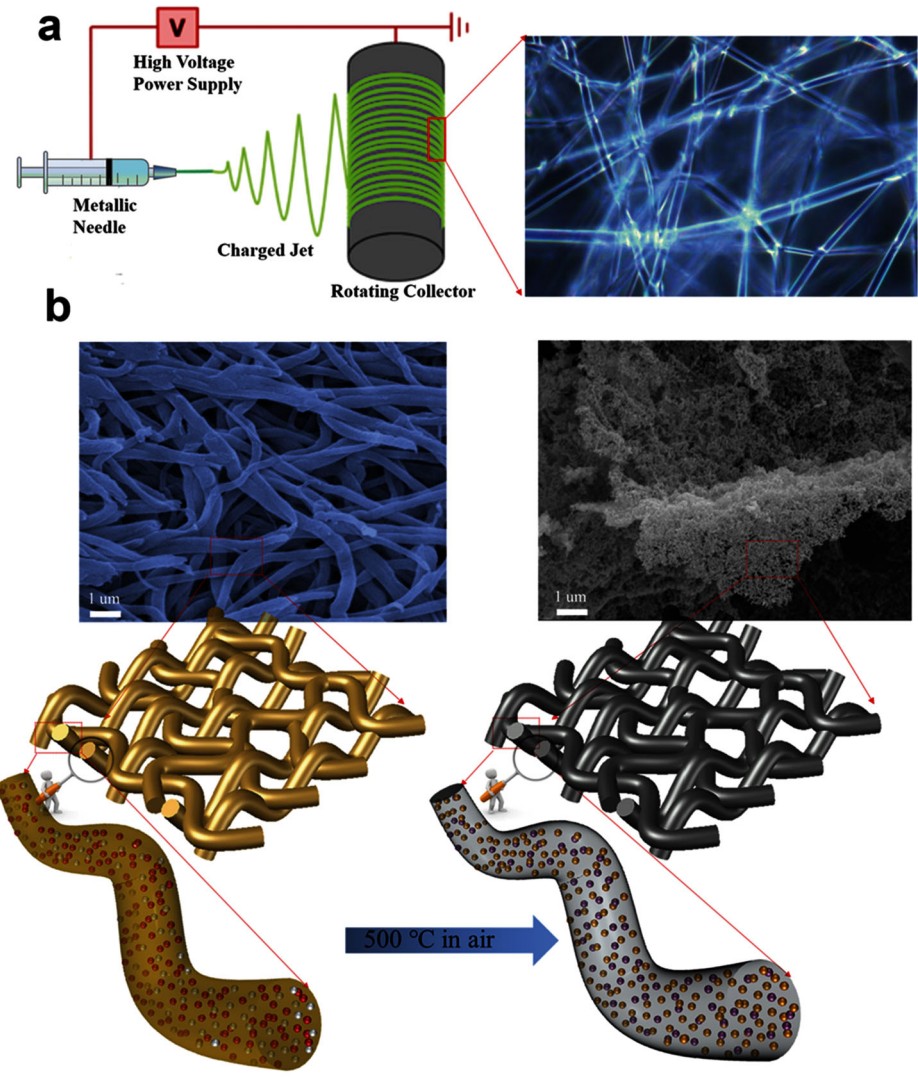

**Fig. 1 Synthetic scheme of metal-semiconductor encapsulated by graphitic carbon layers. a** Diagram of electrospinning setup and the fiber formation in the rotating collector. **b** SEM image of Mo salt, various metal oxides, PVP fibers (left), and metal-metal oxide nanocomposites (right).

further explored using transmission electron microscopy (TEM). The nanorod is connected to each other to form the 3D network heterostructure with the help of graphitic carbon layers (Fig. 2b), consistent with the SEM image (Fig. 1b). Further, high-angle annular dark-field scanning transmission electron microscopy (HAADF-STEM; Fig. S3a) and annular bright-field scanning transmission electron microscopy (ABF-STEM; Fig. S3b) were performed at low-magnification and aided by energy-dispersive X-ray spectroscopy (EDS). The overview morphology and elemental distribution of Ir, Mo, O, and C indicate that the heterostructure of IMO was successfully fabricated. The different contrast TEM images (Fig. 2c, d) show the Mo and Ir distribution according to the atomic number dependence of Z contrast, as verified by the line energy-dispersive X-ray spectroscopy (EDS) elemental maps (Fig. S4). Furthermore, the EDS mapping image proves that the elements of Mo, Ir, and O are distributed homogeneously (Fig. 2e), consistent with Fig. S3.

The high-resolution TEM (HRTEM) image is used to identify the lattice fringes of IMO. Crystalline lattice distances of 0.35 nm and 0.22 nm correspond to the (040) facet of $MoO_3$ and (111) facet of Ir, respectively (Fig. 2g). Note that facets (040) of $MoO_3$

and (111) of Ir can be obtained at 37.8° and 40.7° from the XRD pattern, consistent with the HRTEM images. Also, the lower peak intensity at 40.7° indicated the Ir with crystal size has good dispersion on the $MoO_3$, facilitating the interaction between Ir and $MoO_3$. Note that graphitic carbon layers can be observed in the edge of IMO, evidenced by the contrast HRTEM images (Fig. 2f, g). There is a conception that the graphitic carbon layers are not active sites for the catalytic reaction and only work for increasing the electron transfer and enhancing the catalytic stability[42]. Graphitic carbon layers are not generally considered for further characterization. According to the morphology and structural characterizations of IMO, we proposed the structure model, including the side view and top view of the heterojunction between the $MoO_3$ (040) and Ir (111) facets (Fig. S5). Finally, the atomic resolution ABF-STEM (Fig. 2h) and HAADF-STEM (Fig. S6) images (the Ir metal area in the red box in Fig. 2g) were collected.

**Spectroscopic characterization**. To probe surface compositions and determine the oxidation states, we performed X-ray photoelectron spectroscopy (XPS) analyses. The survey spectra of IMO

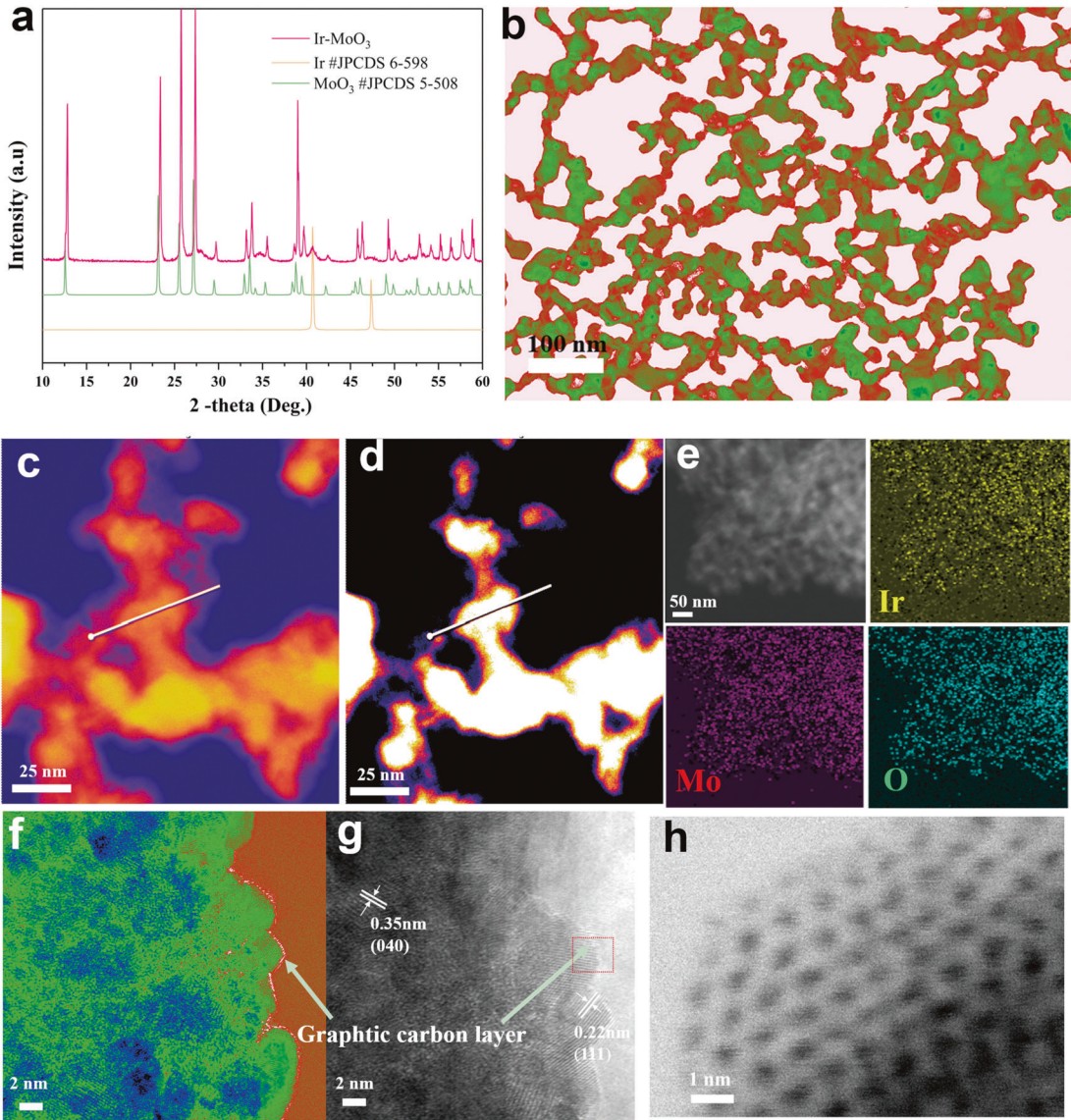

**Fig. 2 Morphology and structural characterizations of IMO. a** PXRD pattern. **b** TEM image. **c, d** TEM images with line EDS by different contrast. **e** TEM-EDS elemental mapping images. **f, g** HRTEM image. **h** ABF-STEM image.

and IOMO show the existence of C, Ir, Mo, and O elements, respectively (Fig. S7), consistent with EDS analysis (Fig. S3). To cross-valid our experimental result, the commercial Ir, IrCl$_3$, and IrO$_2$ are performed, and the detailed fitting parameter was shown in Table S1. Note that all XPS spectra were calibrated using C 1s at 284.6 eV (Fig. S7). The Ir 4f of IMO has a surprisingly high energy shift compared with that of IOMO, indicating the Ir nanoparticles (NPs) have released the electron due to the low electronegativity (Fig. 3a). To gain insight into this phenomenon, we further analyzed the high-resolution XPS (HR-XPS) of Mo 3d and O 1s (Fig. 3b and Fig. S8). While the Mo$^{5+}$ 3d peak is observed clearly from the Mo 3d of IMO, the peak of Mo$^{5+}$ 3d (~231.72 eV) is hardly obtained in IOMO (Fig. 3b). Note that O 1s show traditional three peaks that O$_L$ is lattice oxygen, O$_{OH}$ is adsorbed hydroxide, and O$_{Water}$ is adsorbed water (Fig. S8), consistent with previous reports[44,45]. To further gain insight into the phenomenon, we calculated the charge density difference for investigating the electron transfer over the Ir and MoO$_3$ in IMO (Fig. 3c, d and Fig. S9). Note that the electrons of Ir are transferred to Mo, as observed from the images of charge density difference. We confirm that the Ir NPs of the IMO has an

electron-deficient surface, as evidenced by a higher Ir surface valence state of IMO than that of IOMO and commercial IrO$_2$ (Fig. 3a and Fig. S10). To prove whether the interfacial Ir atoms binding to MoO$_3$ or the surface Ir atoms binding to adsorbed oxygen species have higher valence states, the HR-XPS measurement about Ir 4f in IMO is performed with an argon ion etching treated sample[42], demonstrating zero-valence state of Ir metal in IMO, which is consistent with commercial Ir metal (Fig. S11) and XRD pattern (Fig. 2a). This result confirms that the surface Ir in IMO has a high valence state.

To gain insight into the phase structures of IMO and IOMO, we further utilized the X-ray absorption near-edge structure (XANES) and extended X-ray absorption fine structure (EXAFS) analyses for their bulk-average information. The IMO's Ir L$_3$-edge absorption edge position is lower than that of IrO$_2$ (Fig. 4a), indicating metallic Ir NPs properties, consistent with IMO's XRD pattern (Fig. 2a). Note that Ir L$_3$-edge of IMO is also positively shifted compared with that of Ir metal foil due to the high electron-deficient surface of IMO (Fig. 4a). Moreover, the peak of Ir L$_3$-edge derivative XANES of IMO (Fig. 4b) is located at a low energy position compared with IOMO, implying the metallic

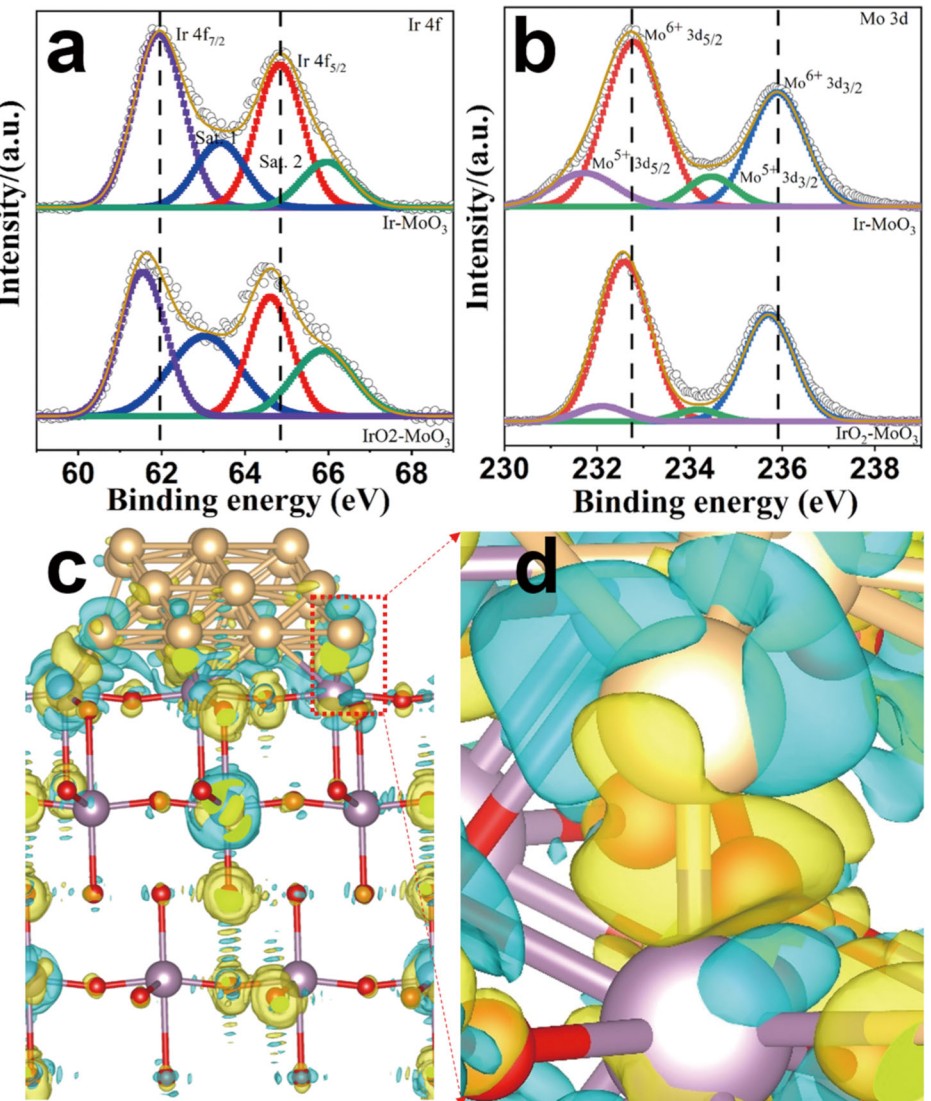

**Fig. 3 Electron transfer behaviors characterizations of IMO. a** HR-XPS of Ir 4*f*. **b** HR-XPS of Mo 3d. **c** The charge density difference of IMO. **d**, The enlarged charge density difference for c. The yellow and cyan regions represent electron accumulation and depletion, respectively. The red, gold, and violet balls represent the O, Ir, and Mo atoms, respectively. The isosurface value is 0.015e/bohr$^3$.

properties of Ir NPs are consistent with XANES and XRD analysis. Fourier-transformed (FT) $k^2$-weighted EXAFS spectra of Ir $L_3$-edge show the prominent peak at 1.65 Å assigned to the Ir-O first coordination shell of $IrO_2$ and the prominent peak at 2.58 Å assigned to the Ir-Ir first coordination shell of Ir NPs (Fig. 4c), which is consistent with previous $IrO_2$[46] and metallic Ir[47]. To cross-validate the IMO's electron-deficient surface, Mo K-edge absorption edge position change (Fig. 4d) was examined as identified by XPS, confirming the electron transfer from Ir element to Mo element. The Mo K-edge absorption of IMO has a shift to low energy, compared to IOMO and $MoO_3$ (Enlarged Fig. 4d), consistent with the $Mo^{5+}$ 3d peak from XPS, although no noticeable phase change of $MoO_3$ was identified from XRD. Correspondingly, the peak derivative of XANES for Mo K-edge in IMO also shifts to low energy as indicated by the red arrow direction and in comparison to IOMO and $MoO_3$ (Fig. 4e), consistent with Ir L3-edge analysis. Note that the difference between IMO, IOMO, and $MoO_3$ is hardly observed from FT $k^3$-weighted EXAFS spectra of Mo K-edge (Fig. 4f), demonstrating the characteristic peaks of $MoO_3$, which is consistent with previous reports[48,49]. Generally, it is hard to observe the

heterointerface bonding for the heterostructure of IMO from EXAFS spectra[50]. Thus, EXAFS fitting curves for IMO and IOMO were conducted to study further the coordination information (Fig. S12 and Table S2), indicating Ir-Mo scattering length is around 2.0 Å, consistent with DFT results.

Wavelet transforms (WT) for the Ir $L_3$-edge and Mo K-edge EXAFS analysis are applied to demonstrate the atomic dispersion of IMO and IOMO, respectively (Fig. 4g). The WT of Ir $L_3$-edge related to Ir–Ir bond is detected in Ir-Foil and IMO, confirming the metallic properties of Ir in IMO. In comparison with IMO, as we expected, IOMO has a WT pattern similar with $IrO_2$, consistent with XRD, XANES, EXAFS analyses. Moreover, the WT patterns for IMO and IOMO exhibited a signal similar to that of $MoO_3$, which differs from Mo-Foil. By combining the XAFS results (Ir $L_3$-edge and Mo K-edge) and XPS data analysis, metallic Ir NPs in IMO have unambiguously higher surface oxidation than $IrO_2$ in IOMO. To clarify and expound on the phenomenon, two critical factors of (i) surface oxygen of Ir evidenced from Argon etching XPS (Fig. S11) and (ii) the electron-withdrawing material of $MoO_3$ documented by charge density difference (Fig. 3c) and XAFS spectra (Fig. 4a, d)

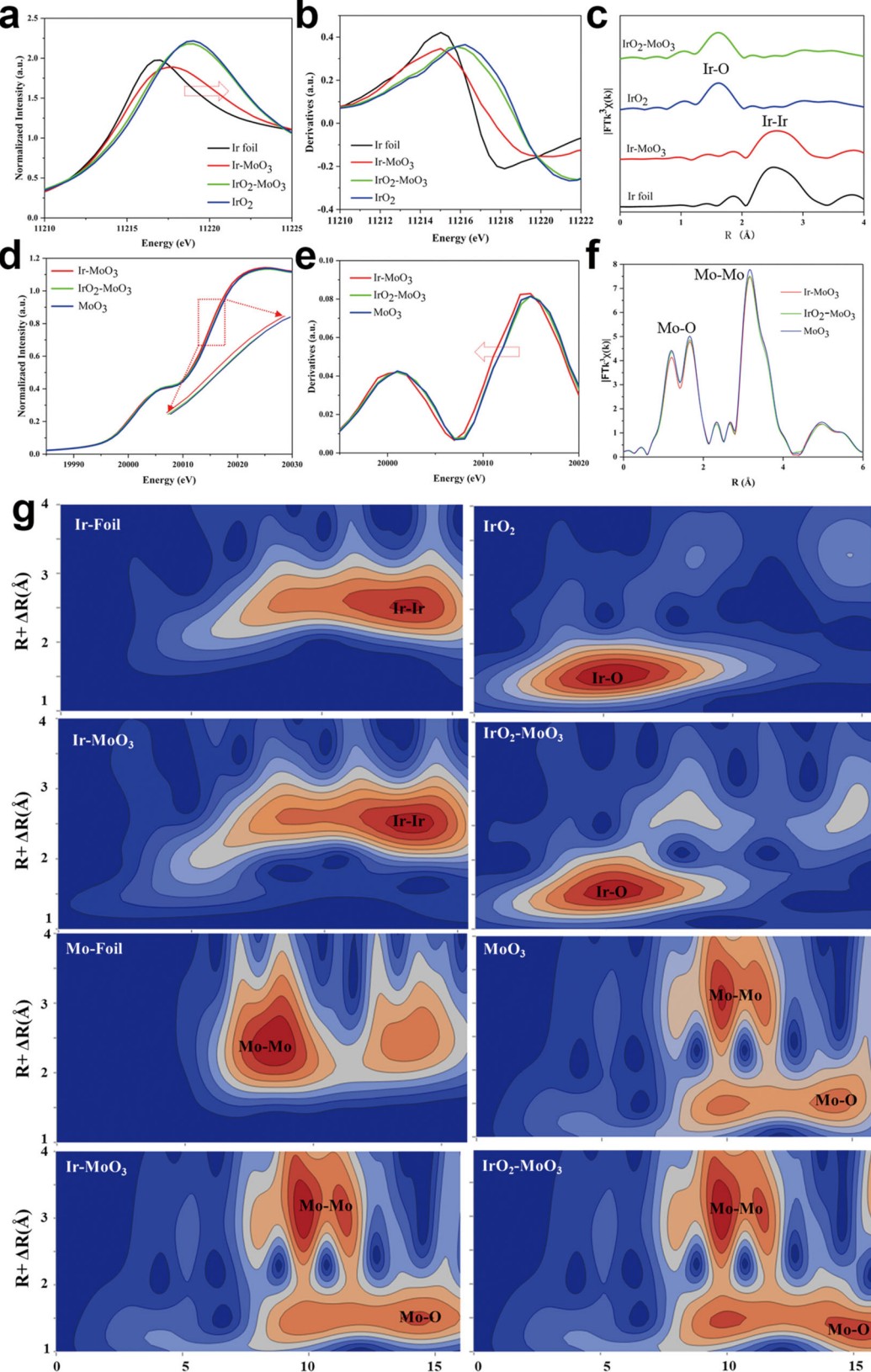

**Fig. 4 X-ray absorption fine structure (XAFS) characterizations. a** XANES survey spectra at the Ir L$_3$-edge for Ir foil, Ir-MoO$_3$, IrO$_2$-MoO$_3$, and IrO$_2$.
**b** Derivative of L$_3$-edge XANES spectra for Ir foil, Ir-MoO$_3$, IrO$_2$-MoO$_3$, and IrO$_2$. **c** Ir L$_3$-edge X-ray absorption Fourier-transformed (FT) $k^3$-weighted
EXAFS spectra in R space. **d** XANES survey spectra at the Mo K-edge XANES spectra for Ir-MoO$_3$, IrO$_2$-MoO$_3$, and MoO$_3$. **e** Derivative of L$_3$-edge XANES
spectra for Ir-MoO$_3$, IrO$_2$-MoO$_3$, and MoO$_3$. **f** Mo K-edge X-ray absorption Fourier-transformed (FT) $k^3$-weighted EXAFS spectra in R space. **g** Wavelet
transforms (WT-EXAFS) for Ir-MoO$_3$, IrO$_2$-MoO$_3$, IrO$_2$, MoO$_3$, Ir foil, and Mo foil.

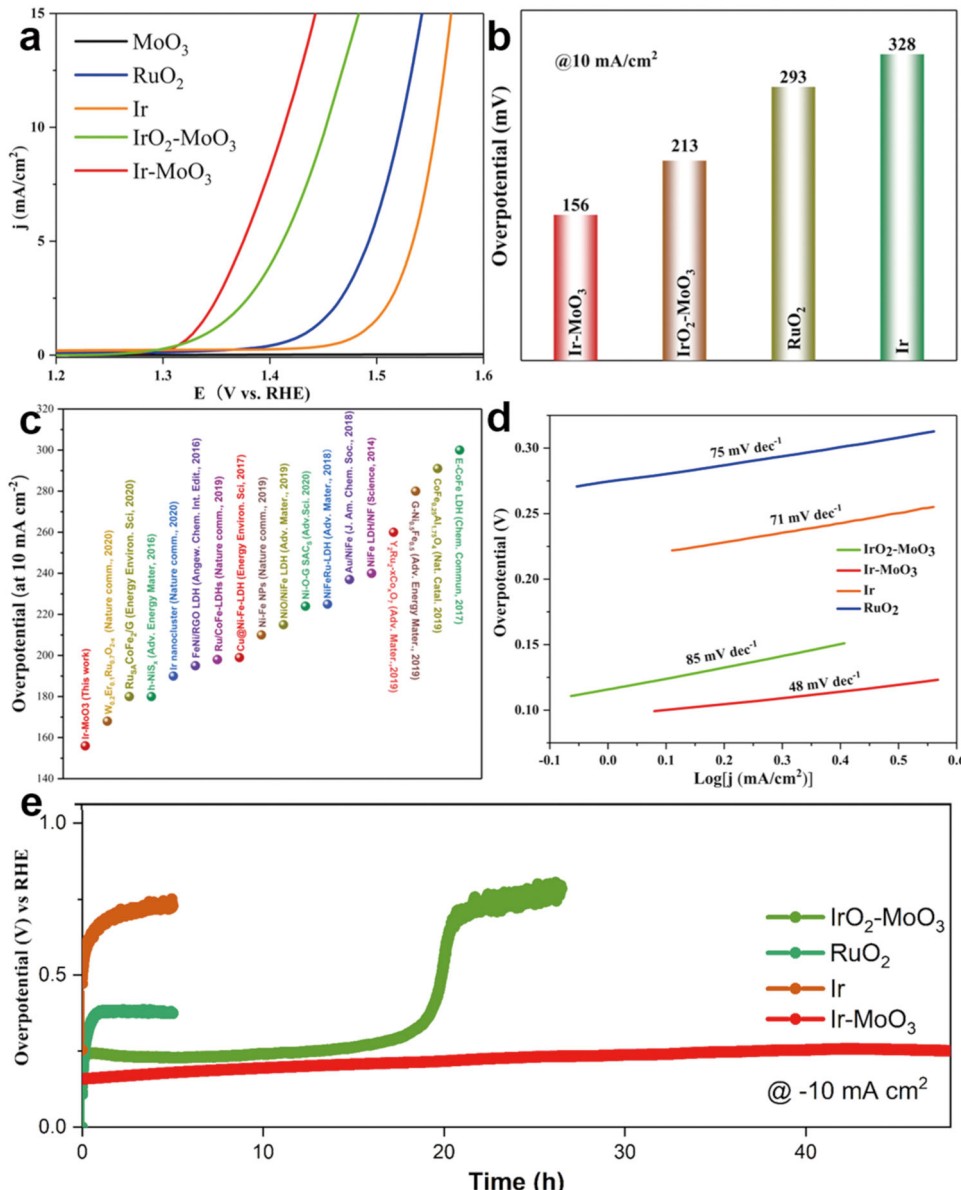

**Fig. 5 Electrochemical oxygen evolution reaction (OER) performance. a** OER polarization curves of as-prepared samples. **b** Comparison of the overpotentials required to reach a current density of 10 mA cm$^{-2}$ **c** Comparison of intrinsic catalytic activities of OER using the literature. **d** Tafel plots of as-prepared electrodes. **e** Chronopotentiometric curves of IrO$_2$-MoO$_3$, Ir-MoO$_3$, RuO$_2$, and Ir.

are predominant. The surface oxygen on the Ir NPs can be attributed to the annealing process in the air, which positively promotes the OER activity[42]. Overall, the IMO has an electron-deficient surface for the two reasons: (1) surface oxygen-group on IMO induced the electron transfer from Ir to O; (2) the MoO$_3$ withdraws the electron from Ir NPs, which is the likely center for the formation of oxygen because of a nucleophilic attack by OH or because of H$_2$O forming an O–O bond[30].

**Electrochemical OER performance**. IMO outperforms other OER catalysts previously reported in the literature, including benchmark catalysts such as RuO$_2$ (10 mA cm$^{-2}$ at ~293 mV, similar to the previous studies[42,51]) and Ir (10 mA cm$^{-2}$ at ~328 mV, similar to the previous studies[51]). The OER catalytic activity of IMO is significantly higher, as evidenced by the overpotential ($\eta$) of ~156 mV under the same current density (10 mA cm$^{-2}$, Fig. 5a, b). Remarkably, due to the unique electron-deficient surface structures (Fig. 5c), IMO exhibits the

best OER efficiency with the ultra-low $\eta$ as compared to recently reported literature. Tafel slopes are derived from the polarization curves to provide deeper insights into the OER mechanism (Fig. 5d). The Tafel slope of IMO is 48 mV dec$^{-1}$, which is remarkably lower than the state-of-the-art RuO$_2$ at 75 mV dec$^{-1}$, consistent with the previous reports[42,51] and Ir at 71 mV dec$^{-1}$, consistent with the previous reports[51]. All measured values of Tafel slope is lower than 120 mV dec$^{-1}$ and we can conclude surface species formed in the step just before the rate-determining step is not predominant[52]. Due to the high coverage of active species at empty sites that decrease the value of Tafel slope[52–54], the low Tafel slope of IMO can be attributed to a large number of oxygen species on surface Ir in IMO. The absorbed oxygen species not only provide the high valence state surface of Ir in IMO but also reduce the value of Tafel slope, which accelerates the process of OER and increases the OER efficiency. The fast electron transfer kinetics process during the OER reaction is consistent with the metallic nature of Ir NPs with graphitic carbon

support. The current density differences show that the double-layer capacitance ($C_{DL}$) is a function of the scan rate (Fig. S13). As a result, the $C_{DL}$ of IMO at ~8.0 mF cm$^{-2}$ is higher than that of IOMO at ~6.4 mF cm$^{-2}$, indicating IMO has a high total electrochemical active surface area, which facilitates the electrochemical process of OER. Nyquist plots of IMO (Fig. S14) show a low electrolyte resistance (Rs) and charge transfer resistance (Rct) compared to IOMO catalysts, implying a fast ion transfer and quickly electron processes at the catalyst–electrolyte interface, consistent with the fast electron transfer kinetics process identified by Tafel slope. Faradaic efficiency of IMO is estimated by measuring the produced $O_2$ gas using a water displacement method[50,55], demonstrating nearly 100% Faradaic efficiency (Fig. S15 and Table S3). To assess the intrinsic catalytic efficiency, we first perform the mass activity based on the total deposited catalysts amount, indicating the superior mass activity of IMO (Fig. S16a). To further quantify the catalytic efficiency of Ir metal, mass activity is evaluated by normalizing the amount of Ir. IMO delivers a high mass activity of ~178.9 mA/mg$_{ir}$ at an overpotential of 200 mV, which is about two times higher than that of IOMO and about 600 times higher than that of Ir metal (Fig. S16b). Furthermore, turnover frequency (TOF) is performed (Fig. S17), consistent with the result of mass activity. Remarkably, the TOF of IMO reaches 5.2 s$^{-1}$ at an overpotential of 200 mV, 40 times higher than that of Ir metal, indicating its excellent intrinsic activity due to the highly electron-deficient surface of Ir metal in IMO, which is promising for practical usage.

To evaluate the catalysts' durability, the chronopotentiometry curves of IMO, Ir, and $RuO_2$ were collected over 48 h. For IMO, no tremendous loss occurred at the constant anodic current density of 10 mA cm$^{-2}$ under 0.5 M $H_2SO_4$. By comparison, Ir and $RuO_2$ were found to be unstable with $\eta$ increasing fast within few hours (Fig. 5e). Further, the chronopotentiometry test at a high current density of 100 mA cm$^{-2}$ for 48 h demonstrates the exceptional stability of IMO, which is promising for practical usage (Fig. S18). Note that the activity and stability generally show the trade-off relationship. Thanks to the $Mo^{5+}$ existence in the $MoO_3$, the catalysts of IMO exhibited excellent durability compared with IOMO. To gain insight into the mechanism, we measure the XPS spectra after the OER experiment. The energy state of the Ir, Mo, and O undergo a minor high-energy shift after the OER process, indicating that these elements lose the electron during the OER (Fig. S19 and Table S4). Note that the surface structure of the IMO is still preserved well after the OER reaction, since the synergic effect of the high surface state of Ir with the help of the $Mo^{5+}$ can withstand resistance in an oxidation state (Fig. S19a). However, the surface structure of IOMO can be corroded during the OER process because the ratio of lattice oxygen ($O_L$) decreases compared with that in fresh of IOMO (Fig. S19b), which is the origin of the instability for IOMO in comparison with IMO.

**Theoretical investigation for active sites using DFT.** Inspired by the unique structure with the electron-deficient surface that benefits the OER efficiency as demonstrated using in situ and ex situ x-ray spectroscopies[30], we further investigate the catalytic activity by means of DFT. Firstly, we check the favorite sites for HO* adsorption at various IMO sites since the HO* is the first intermediate formed in the OER (Fig. 6a). The interfacial site (A site) between the $MoO_3$ and Ir, the Ir edge site (B site), the Ir hollow site (C site), and the Ir top site (D site) are calculated as favorable for adsorption energy of HO*, demonstrating the sites A, B, and C are favorable for the HO* adsorption compared with the D site (Fig. 6b). Note that the D site on IMO is unstable during the optimized process, which prevents the D site from

becoming the active site for the OER process. Additionally, note that the same trend of oxygen adsorption is observed and the D site on IMO shows the bond distance of 2.72 Å (Fig. S20), indicating that the D site is not a favorable site for oxygen adsorption. Thus, we adopt sites A, B, and C to further study mechanism of site activity. With its electron-deficient surface, IMO has a high OER catalytic efficiency and fast kinetic processes, resulting in the boosted OER performance[30,42]. The B site is more favored for the OER process compared to sites A and C because the configuration of HOO* cleaves as O* and HO* during the optimization process at sites of A and C sites (Fig. 6d–f). Although the intermediate configuration of HOO* at site B is stable (Fig. 6c, d and Fig. S21), but the energy barrier is still high.

To gain insight into the OER mechanism of IMO by considering its boosting OER efficiency, the various amount of surface oxygen species was considered (Figs. S22–26). As a result, the Ir metallic surface with 8 surface oxygen atoms has the highest formation energy (−1.74 eV/atom) compared to other amount surface oxygen atoms (−1.63 eV/atom for seven surface oxygen; −1.62 eV/atom for nine surface oxygen and −1.42 eV/atom for ten surface oxygen). As we expected, the IMO with seven surface oxygen atoms (O-7) models shows the low OER energy barriers by reducing the energy barriers compared with the IMO (Fig. 6c and Fig. S27). In addition, to break the scaling relation between HOO* and HO*[14,56], the proton dissociation pathway (PDP) is suggested for the models IMO (O-8). As a result, PDP indicates the lowest energy barrier when the proton transfers to the neighbor surface oxygen compared to the relative other OER pathway, and the corresponding configurations are illustrated (Fig. 6c), which is consistent with the experimental results of low overpotential and Tafel slope. These results significantly demonstrate that the surface oxygen participates in the OER reaction as a proton acceptor, powerfully uncovering the origin of IMO's excellent catalytic OER performance, which opens up an avenue for designing highly efficient catalysts.

By following data-driven and high-throughput simulation guidance for the coupled Ir-Mo system as a promising acid-stable catalyst, we designed and fabricated the electron-deficient surface of IMO catalysts using a pot economy and one-pot electrospinning strategy. Ir 4f of IMO has a high energy shift from the HR-XPS measurement compared to that of IOMO due to the surface oxygen and electron-withdrawing substrate, as cross-validated from the DFT simulation (charge density difference) and XAFS spectra. To our knowledge, as a result, IMO shows the high stability with superior OER catalytic performance compared with the benchmark catalysts (Ir and $RuO_2$) up to date, because the synergic effect of the high surface state of Ir with the help of the $Mo^{5+}$ can withstand resistance in an oxidation state. Further, the active catalytic origin of OER was uncovered by means of DFT simulation. The proton dissociation mechanism was suggested while the neighbor surface oxygen works as a proton acceptor. This study not only uncovers the rational design of IMO for superior catalytic performance by creating an electron-deficient surface, but also reveal the general and unique strategy—PVP with following annealing process to fabricate the metal–metal oxide (Ir-$MoO_3$, Ru-$MoO_3$, Rh-$MoO_3$, and Au-$MoO_3$) heterostructures—for guiding other metal-semiconductor design.

## Methods

**Computational details.** All our DFT calculations were carried out using the Vienna Ab initio Simulation Package (VASP)[57–59]. The projector-augmented wave (PAW) method was employed to describe the ion-electron interaction[60,61]. The exchange-correlation function was depicted by the generalized gradient approximation in the form of Perdew–Burke–Ernzerhof (GGA-PBE) functional[62]. The electronic wave functions were expanded using a plane-wave basis with 400 eV cutoff energy. Monkhorst-Pack $k$ points mesh sizes of $2 \times 2 \times 1$ were used for

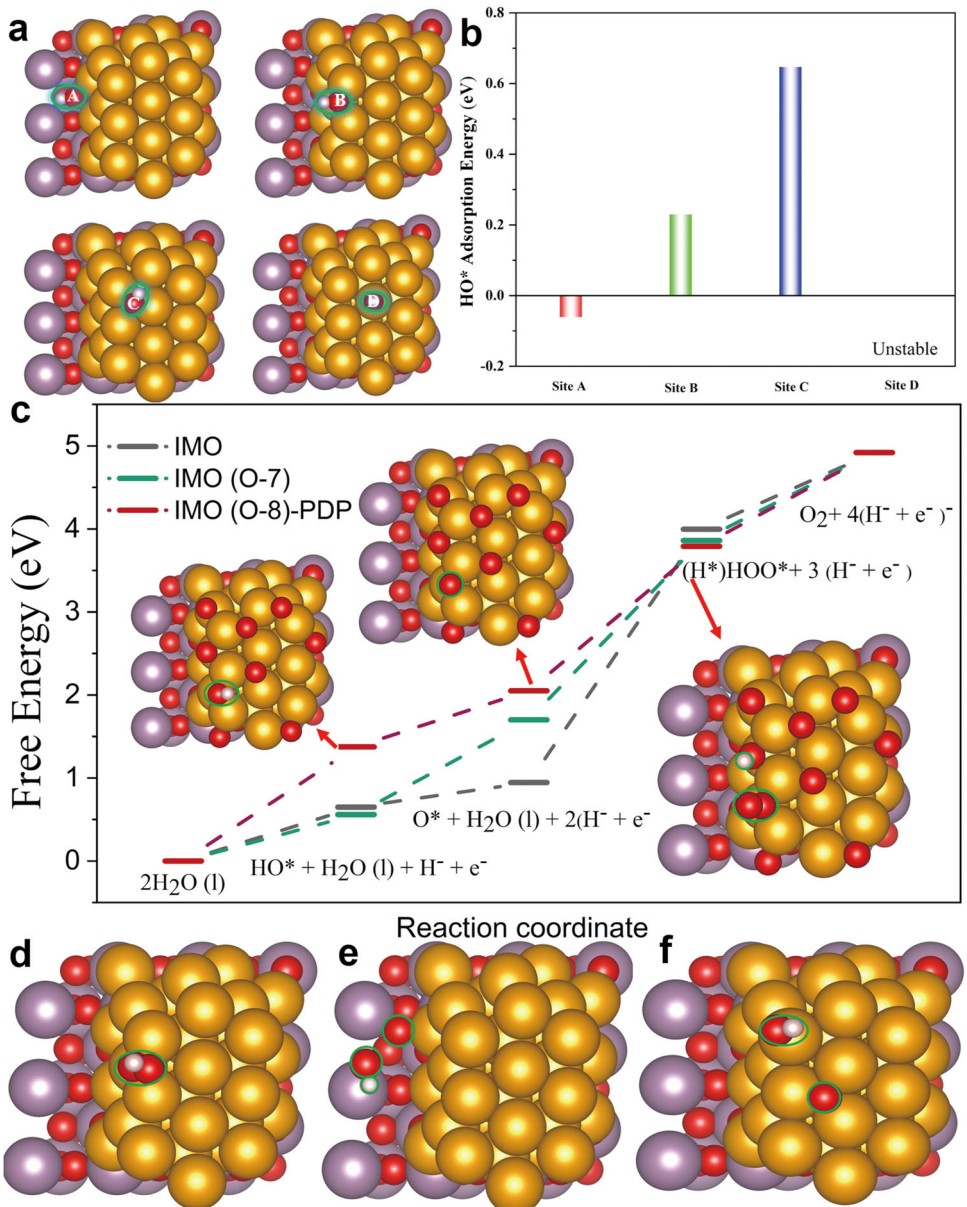

**Fig. 6 DFT computational modeling and investigation of IMO active site. a** The optimal sites for the HO* adsorption on the interfacial site (A site) between the MoO₃ and Ir, Ir's edge site (B site), Ir's hollow site (C site), and Ir's top site (D site). **b** The corresponding adsorption energies of the four sites in a. **c** Relative energy profiles, and the simplified surface structures of various reaction species as the arrow define. **d** The configuration of HOO* for site B after relaxation. **e** The configuration of HOO* for site A after relaxation. **f** The configuration of HOO* for site C after relaxation. Color code: The red, gold, and violet balls represent the O, Ir, and Mo atoms, respectively.

geometric optimizations and self-consistent total energy calculations[63,64]. All the structures were optimized with the convergence criteria of $10^{-4}$ eV/Å in force and $10^{-6}$ eV per atom in energy. A 15 Å vacuum space in the z-direction was applied to avoid interactions between adjacent periodic images.

The thermodynamic model of water oxidation proposed by Norskov and co-workers[65], which is composed of four electrochemical steps, each of which constitutes one proton transfer, was used in this work. The following electron reaction paths are considered for the oxygen evolution reaction (OER) process:

$$H_2O(l) + * \rightleftharpoons HO* + H^+ + e^- (\triangle G_1), \quad (1)$$

$$HO* \rightleftharpoons O* + H^+ + e^- (\triangle G_2), \quad (2)$$

$$O* + H_2O(l) \rightleftharpoons (H*)HOO* + H^+ + e^- (\triangle G_3), \quad (3)$$

$$HOO* \rightleftharpoons O_2(g) + * + H^+ + e^- (\triangle G_4), \quad (4)$$

where * represents an active site on the Ir-MoO₃ surface. O*, OH*, and OOH* are intermediates adsorbed on the active site.

Molecular O₂ energy is indirectly obtained from the experimental Gibbs free energy formation of water oxidation equation at standard conditions $2H_2O \rightarrow O_2 + 2H_2$, $\triangle G = -4.92$ eV to avoid the calculation of oxygen molecule O₂ by DFT[66]. For the ideal case in which $\triangle G_1 = \triangle G_2 = \triangle G_3 = \triangle G_4 = 0$, the equilibrium potential to produce oxygen is $(eU - k_B T \ln(10) \times pH)/e = 1.23$ V. Practically, the catalytic activity of the above process is controlled by the rate-determining step (RDS), which has the maximum free energies of adsorption ($\triangle G_{1-4}$).

**Synthesis of MoO₃ composite**. MoO₃ was prepared by means of a facile electrospinning method in the presence of ammonium molybdate (para) tetrahydrate, and polyvinylpyrrolidone (1.2 g, PVP, average M.W. 130,000), followed by high-temperature pyrolysis. Specifically, 1 mmol ammonium molybdate (para) tetrahydrate and 1.2 g of PVP were added to a mixed solvent of dimethylformamide (5 mL DMF) and alcohol (5 mL EtOH) with stirring at 80-degree for 12 h. The mixed solution was added into a 10 mL plastic syringe with a 25-gauge stainless-steel needle. The applied direct current voltage and distance between the needle and the aluminum-foil-wrapped collector were fixed at 15 kV and 12 cm, respectively. The flow rate of the mixed solution was maintained at 10 μL min⁻¹. The as-

prepared materials were annealed at 500 °C for 3 h in the air at a heating rate of 5 °C min$^{-1}$. Finally, MO$_3$ was collected.

**Synthesis of Ir and MoO$_3$ composite (IMO)**. IMO was prepared by means of a facile electrospinning method in the presence of IrO$_2$, ammonium molybdate (para) tetrahydrate, and polyvinylpyrrolidone (1.2 g, PVP, average M.W. 130,000), followed by high-temperature pyrolysis. Specifically, 1 mmol IrO$_2$, 1 mmol ammonium molybdate (para) tetrahydrate, and 1.2 g of PVP were added to a mixed solvent of dimethylformamide (5 mL DMF) and alcohol (5 mL EtOH) with stirring at 80-degree for 12 h. The mixed solution was added into a 10 mL plastic syringe with a 25-gauge stainless-steel needle. The applied direct current voltage and distance between the needle and the aluminum-foil-wrapped collector were fixed at 15 kV and 12 cm, respectively. The flow rate of the mixed solution was maintained at 10 μL min$^{-1}$. The as-prepared materials were annealed at 500 °C for 3 h in the air at a heating rate of 5 °C min$^{-1}$. Finally, IMO was collected. Note that the Ru-MoO$_3$, Rh-MoO$_3$, and Au-MoO$_3$ were prepared by the same method with the replacement of the appropriate metal oxides.

**Synthesis of IrO$_2$ and MoO$_3$ composite (IOMO)**. IOMO was prepared by means of a facile electrospinning method in the presence of ammonium molybdate (para) tetrahydrate, and polyvinylpyrrolidone (1.2 g, PVP, average M.W. 130,000), followed by high-temperature pyrolysis. Specifically, 1 mmol ammonium molybdate (para) tetrahydrate and 1.2 g of PVP were added to a mixed solvent of dimethyl-formamide (5 mL DMF) and alcohol (5 mL EtOH) with stirring at 80-degree for 12 h. The mixed solution was added into a 10 mL plastic syringe with a 25-gauge stainless-steel needle. The applied direct current voltage and distance between the needle and the aluminum-foil-wrapped collector were fixed at 15 kV and 12 cm, respectively. The flow rate of the mixed solution was maintained at 10 μL min$^{-1}$. The as-prepared materials were mixed with IrO$_2$ (1 mmol) and then were annealed at 500 °C for 3 h in the air at a heating rate of 5 °C min$^{-1}$. Finally, IOMO was collected.

**Electrochemical measurements**. All electrochemical tests were carried out using a VMP3 electrochemical workstation (Biologic Science Instruments, France). To prepare the working electrode, we sonicated a mixture of 5 mg of catalyst, 0.5 mL of DI water, 0.49 mL of ethanol, and 10 μL of 5 wt% Nafion for 20 min. Then ink was drop-cast onto carbon paper to yield a weight density equivalent to $z$.

Electrocatalytic OER activity of as-prepared electrodes was assessed in an Ar-saturated 0.5 M H$_2$SO$_4$ solution using a standard three-electrode cell. All potentials of these samples were referenced to the reversible hydrogen electrode (RHE).

Ag/AgCl (3 M KCl solution) was used as the reference electrode, platinum foil was used as the counter electrode, and each of the synthesized materials was used as a working electrode. Linear sweep voltammograms (LSV) were acquired to assess HER performance; all LSVs were acquired at the scan rate of 5 mV s$^{-1}$. Potentials were converted to the RHE scale using the equation $E_{RHE} = E_{Ag/AgCl} + E^0_{Ag/AgCl} + 0.059 \times pH$, where $E_{RHE}$ is the potential relative RHE, $E_{Ag/AgCl}$ is the potential measured potential against an Ag/AgCl reference electrode, $E^0_{Ag/AgCl}$ is the standard potential of Ag/AgCl reference electrode at room temperature. All LSV polarization curves were corrected for ohmic drops in solution.

**Materials characterizations**. Sample morphologies were investigated using a JSM 7401F (JEOL Ltd., Tokyo, Japan) scanning electron microscope (SEM) operated at 3.0 kV. High-resolution TEM (HRTEM) images were collected on a JEOL ARM-2100F field-emission transmission electron microscope operated at 200 kV accelerating voltage. X-ray photoelectron spectroscopic (XPS) measurements were performed on a VG Microtech ESCA 2000 using a monochromic Al X-ray source (97.9 W, 93.9 eV); Note that all XPS spectra were calibrated using C 1$s$ at 284.6 eV. X-ray diffraction (XRD) patterns were recorded using an Ultima IV (Rigaku) instrument using substrates directly. Ir L$_3$-edge and Mo K-edge XAFS experiments were conducted in an ambient condition at 10C beamline of Pohang Light Sources-II (PLS-II). The incident beam was monochromatized using a Si (111) double crystal monochromator and detuned to ~70% of its maximum intensity for reducing higher-order harmonics. For energy calibration, the first inflection points of reference metal foils were set to the corresponding absorption edge energy (11.2152 keV for Ir-L3 edge and 19.995 keV for Mo K-edge). After normalization, the EXAFS signal was $k^3$-weighted to magnify high-energy oscillations and Fourier-transformed in the $k$-range from 2.5 to 11.5 Å$^{-1}$.

## Data availability
The data supporting the findings of this study are available within the paper and its Supplementary Materials. The raw data used in this work are available from the corresponding author upon reasonable request.

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

## Acknowledgements

This work was supported in Korea by the Institute for Basic Science (IBS-R011-D1) and partially by the Technology Development Program of MSS [S2980892] & the ICT development R&D program of MSIT [S2980892], and the INNOPOLIS Foundation (2020-DD-SB-0731). X.L. thanks Prof. Zhongfang Chen for the constructive suggestion. X.L. thanks Mallikarjuna Reddy Kesama for the design of the figures of the diagram.

## Author contributions

X.L and H.L. conceived the idea and directed the project. X.L. design and synthesize the experiments, analyzed data, and performed DFT studies. H.K., and S.X. help with experimental synthesis, XAFS and analysis. A.K. help the OER performance measurement. N.Q.T. help the Ir and RuO$_2$ samples. J.W. give constructive comments. J.L., and T.Y. help the SEM measurement. X.S. help the XPS analysis. M.L. help the XRD analysis. M.G.K. performed the XAFS measurement. X.L. and H.L. write the draft. All authors discuss the results.

## Competing interests

The authors declare no competing interests.
