## [Peer Review File · Nature Communications]

Restructuring highly electron-deficient metal-metal oxides for boosting stability in acidic oxygen evolution reactionREVIEWER COMMENTS

Reviewer #1 (Remarks to the Author):

The manuscript entitled "Restructuring highly electron-deficient metal-metal oxides for boosting stability in acidic oxygen evolution reaction" has reported highly electron-deficient metal on semiconducting metal oxides as a high-performance catalyst for OER catalysis. The conception that highly electron-deficient surface of metal can contribute to the outstanding OER performance is novel and interesting. Meanwhile, the whole paper is well written and the characterizations can basically support the conclusions, thus, I could recommend it to be published on Nature Communications. However, some major problems still need to be addressed.

1. Why the valence state of surface atom of Ir NPs can exceed that of Ir atom in IrO₂ whose Ir atoms are even surrounded by O atoms? In addition, if a highly electron-deficient metal can support excellent OER performance, why not select the IrO₃ that has Ir atoms bearing 6+?
2. Which atoms (or part) of Ir NPs in IMO show valence state higher than 4+, the interfacial Ir atoms binding to Mo? Or the Ir atoms binding to adsorbed oxygen species? and why? This issue is critical for active sites in OER and need to be explained in the work.
3. Since the part of Ir NPs in IMO is oxidized to high valence state, the Ir L3-edge may also positively shift compared with the Ir metal, like the results of Mo K-edge in IMO. Hence, a comparison of Ir L3-edge between IMO and Ir metal is needed in this work.
4. For practical usage, a durability test under higher current density (e. g. >50 mA/cm²) is recommended to be provided for comparison in the work.
5. In the part of choosing active sites for OER, the OH* should be a more proper adsorption species for testing rather than O₂, because the OH* is the first intermediate formed in OER.
6. Some relevant works are recommended to be cited: Advanced Functional Materials, 2020, 1908708; Nature Communications, 2021, 12, 2351; Nature Energy, 2019, 4, 329; Advanced Functional Materials, 2018, 1805227

Reviewer #2 (Remarks to the Author):

In this manuscript X. Liu et al. fabricated and characterized an Ir based IMO electrocatalyst for the OER. They found that these catalysts present higher OER activity at low over-potential as well as higher activities under reaction conditions. The manuscript is well written, and the results are well supported. However there are some issues that the authors should consider in order to improve the quality of this manuscript:

- 1) Can the author include some reference for a better comparison with the acquired spectra in this research (i.e. Ir⁰ and IrO₂ (IV))? Ideally one should consider to collect both XPS and NEXAFS.
- 2) For this reviewer is difficult to follow what are Figures 2g and 2f. These images are better explained in Fig. S5. The authors should consider to swap images from the SI to the main text and remove Fig S5 from the SI.
- 3) The EELS spectra are of bad quality. The authors should consider to plot another EELS spectra with better quality or remove these spectra from the text. The information provided by the X-ray spectroscopy should be enough.
- 4) The XPS spectra fitting is not optimal. It is necessary to re-fit the whole set of spectra, indicate the parameters used (peak position, peak shape and fwhm), and justify why these peaks were used to fit the spectra. In this point the reference spectra can help significantly. In addition it will be nice to show the C1s and VB. Furthermore, it should be indicated how the spectra were calibrated.
- 5) Figure 5e should be plotted without the XPS spectra for clarity.

6)I cannot see Figure S14 in the SI.

Reviewer #3 (Remarks to the Author):

The paper of Liu et al. presents synthesis and characterization of metal/MoO₃ catalysts. The materials were prepared using electrospinning synthetic method and analyzed by microscopic and spectroscopic methods, while their oxygen evolution activity and durability were studied in an RDE set-up. The DFT calculations on different IR sites were also provided in this work.

However, this paper has to be reviewed before publishing, specifically the following points has to be addressed:

1)energy calibration in XPS and XANES analysis was not mentioned in the experimental section, while it is very important while discussing the spectra shifts. Are the observed shifts in XPS related to charging (MoO₃ as a semiconductor may lead to this)? In case of XANES, the Ir edge (metal)is at 11.21520 keV, that does not correspond to the metallic Ir provided in the manuscript

2) Analysis of XPS spectra. The comparison of Ir and Mo core-levels cannot be done when the fitting parameters are not consistent (the intensity ratio are not fixed, the relative binding energy etc). Please correct this.

3) The authors claim the formation of metallic Ir in case of IMO sample where there is no spectroscopic evidence, it looks like it is an oxide, however the oxidation state is lower than IV. Moreover, the analysis of Ir4f XPS spectra shows higher binding energy in comparison to IrO₂, that is related to Ir (III) oxidation state as it was discussed in many publications previously (exception to a common situation higher BE = higher oxidation state).

4) The analysis of the electrochemical data shows higher activity of the synthesized materials, specifically for IMO. The Tafel slope of 48 mV dec⁻¹ in case of IMO in comparison to other materials with 70-80 mV dec⁻¹ should correspond to a change in the rds, however nothing is discussed in the manuscript.

5) The durability analysis of IOMO (Figure 5e) shows an interesting change at 20h, what is happening to the sample ? Was it analyzed ? This should be discussed in the manuscript and compared with IMO behavior.

6) Based on the DFT data, the authors proposed the OER mechanism on the synthesized materials, however it is not well discussed and not clear to a reader - has to be improved.

Responses for the Nature Communications.

Manuscript ID: NCOMMS-21-23624-T

Title: Restructuring highly electron-deficient metal-metal oxides for boosting stability in acidic oxygen evolution reaction

We are grateful to the editor, editorial staff, and reviewers for their critical comments and valuable suggestions. The manuscript has been strictly modified and improved after addressing all the suggestions as listed below:

(The explanations to the comments from reviewers are shown in **blue color with yellow highlight**).

Reviewer(s)' Points-by-points responses to Author:

Reviewer #1

Reviewer #1: The manuscript entitled “Restructuring highly electron-deficient metal-metal oxides for boosting stability in acidic oxygen evolution reaction” has reported highly electron-deficient metal on semiconducting metal oxides as a high-performance catalyst for OER catalysis. The conception that highly electron-deficient surface of metal can contribute to the outstanding OER performance is novel and interesting. Meanwhile, the whole paper is well written and the characterizations can basically support the conclusions, thus, I could recommend it to be published on Nature Communications. However, some major problems still need to be addressed.

Response: We are grateful for the time and effort. Reviewer 1 has reviewed our manuscript, which are constructive and valuable to improve our manuscript.

(1-1-1) Why the valance state of surface atom of Ir NPs can exceed that of Ir atom in IrO₂ whose Ir atoms are even surrounded by O atoms? In addition, if a highly electron-deficient metal can support excellent OER performance, why not select the IrO₃ that has Ir atoms bearing 6+?

Answer 1-1-1) Yes, thanks for the constructive comment. As we mention in our manuscript, we are sure that **the valance state of surface atoms of Ir NPs on IMO nanocomposite can exceed the Ir atoms in IrO₂ by two factors: (i) surface oxygen of Ir; (ii) the electron-withdrawing material of MoO₃.** The detailed explanation with simple model (Figure R1) is as below:

Figure R1. The simple demonstrating two factors for making **the highly electron-deficient surface of Ir** in IMO. Note that the Ir binding with the surface oxygen has the high valence state, evidenced from the argon-etching XPS spectra (detailed information was shown below).

- (i) **Surface oxygen**: The surface oxygen can take the electron from the surface of Ir NPs metal, which induces the high valence state of the surface atom of Ir NPs.

- (ii) *Electron-withdrawing material of MoO₃*: Based on the charge density difference (Figure 3c and 3d), we can clearly see that the electrons of Ir NPs are transferred to the electron-withdrawing MoO₃.

Figure 3. **c**, The charge density difference of IMO. **d**, The enlarged charge density difference for Figure 3c. The yellow and cyan regions represent electron accumulation and depletion, respectively. The red, gold, and violet balls represent the O, Ir, and Mo atoms, respectively. The isosurface value is 0.015e/bohr³.

Based on two reasons that can support solid evidence of XPS results of Ir 4f, we concluded that **the valance state of the surface atom of Ir NPs in IMO exceeds that of Ir atom in IrO₂**. In addition, revealing **the Ir atom bound to the oxygen species in IMO** shows a high valance state. To prove our arguments, **we added new XPS data after argon (Ar) etching** as below. To uncover which part of Ir NPs in IMO shows the valance state higher than 4⁺, the high-resolution

XPS measurement was conducted **before and after Ar etching experiment** (*Energy Environ. Sci.*, 2020, 13, 5152-5164). As we expected, the Ir metallic property of Ir NPs in IMO was revealed after argon etching (Figure S11), which is consistent with the commercial Ir metal. **The oxygen adhering to the Ir surface was removed by the Ar etching, and finally Ir metal properties were obtained.** This is consistent with XRD pattern and EXAFS spectra. We newly added Fig. S11 in the revised Supporting Information.

Figure S11. HR-XPS of Ir 4f for the Commercial Ir, IMO, and argon (Ar) etching IMO.

In addition, we newly added a sentence in our revised manuscript as **“To prove whether the interfacial Ir atoms binding to MoO₃ or the surface Ir atoms binding to adsorbed oxygen species**

have higher valence states, the HR-XPS measurement about Ir 4f in IMO is performed with an argon ion etching treated sample⁴¹, demonstrating zero-valence state of Ir metal in IMO, which is consistent with commercial Ir metal (Figure S11) and XRD pattern (Figure 2a). This result confirms that the surface Ir in IMO has a high valence state.”

(1-1-2) “In addition, if a highly electron-deficient metal can support excellent OER performance, why not select the IrO₃ that has Ir atoms bearing 6+?”

Answer 1-1-2) Yes, thanks for the constructive comment. We believe that a highly electron-deficient metal can support excellent OER performance, based on our experimental result and previous literature (*ACS Energy Lett.* 2021, 6, 4, 1588–1595;). However, we did not select IrO₃, based on the below two reasons:

Reason 1: From the Open Quantum Materials Database (OQMD) that was reported in Figure R2 (*ref: npj Comput Mater*, 2015, 1, 15010), we can see clearly that the IrO₃ is thermodynamically unstable in comparison with IrO₂, which is unsuitable for the highly oxidative environment during the OER process.

Figure R2. <http://oqmd.org/materials/composition/Ir-O>. (*npj Comput Mater*, 2015, 1, 15010)

Reason 2: To our best knowledge, S. Stucki and coworkers firstly proposed OER degradation via the formation of volatile iridium species in the IrO_3 state, which is similar to an intermediate of RuO_4 for Ru (*R. Kötzt, H. Neff, S. Stucki, J. Electrochem. Soc.* 1984, 131, 72–77). Specifically, IrO_3 may corrode into electrolyte as IrO_4^- ion, as shown in Figure R3.

Fig. 6. Model for charge storage and oxygen evolution on iridium electrodes.

Figure R3. Model for charge storage and oxygen evolution on iridium electrodes (*R. Kötz, H. Neff, S. Stucki, J. Electrochem. Soc. 1984, 131, 72–77.*).

In addition, recently, the IrO_3 was further reported to be the gaseous and volatile intermediate in the OER process (*Angew. Chem. Int. Ed. 2018, 57, 2488–2491; Chem. Mater. 2019, 31, 5845–5855*). Based on these reasons above, **we could not select the IrO_3 as a catalyst for the stability issue.**

(1-2) Which atoms (or part) of Ir NPs in IMO show valance state higher than 4+, the interfacial Ir atoms binding to Mo? Or the Ir atoms binding to adsorbed oxygen species? and why? This issue is critical for active sites in OER and need to be explained in the work.

Answer 1-2) Thanks for the constructive comment. The part of Ir atoms binding to adsorbed oxygen species in IMO shows a high valance state. To prove our arguments, we added new XPS data after argon (Ar) etching as below. To uncover which part of Ir NPs in IMO shows the valance state higher than 4⁺, the high-resolution XPS measurement was conducted **before and after Ar etching experiment** (*Energy Environ. Sci.*, 2020, 13, 5152-5164). As we expected, **the Ir metallic property of Ir NPs in IMO was revealed after argon etching (Figure S11), which is consistent with the commercial Ir metal.** The oxygen attached to the Ir surface was removed by the Ar etching and finally gave Ir metallic property, which is consistent with XRD and EXAFS spectra. We newly added Fig. S11.

Figure S11. HR-XPS of Ir 4f for the Commercial Ir, IMO, and argon (Ar) etching IMO.

In addition, we newly added a sentence in our revised manuscript as “To prove whether the interfacial Ir atoms binding to MoO₃ or the surface Ir atoms binding to adsorbed oxygen species have higher valence states, the HR-XPS measurement about Ir 4f in IMO is performed with an argon ion etching treated sample⁴¹, demonstrating zero-valence state of Ir metal in IMO, which is consistent with commercial Ir metal (Figure S11) and XRD pattern (Figure 2a). This result confirms that the surface Ir in IMO has a high valence state.”

(1-3) Since the part of Ir NPs in IMO is oxidized to high valence state, the Ir L3-edge may also positively shift compared with the Ir metal, like the results of Mo K-edge in IMO. Hence, a comparison of Ir L3-edge between IMO and Ir metal is needed in this work.

Answer 1-3) Thanks for the constructive comment. We totally agree review comments. We added the comparison of Ir L3-edge between IMO and Ir metal, as shown revised Figs. 4a and 4b by adding Ir foil. As we expected, the **Ir L3-edge (red color, Ir-MoO₃) was also positively shifted** compared with the Ir metallic foil (black color), consistent with the results of Mo K-edge in IMO.

Fig. 4 | X-ray absorption fine structure (XAFS) characterizations. a, XANES survey spectra at the Ir L₃-edge for Ir foil, Ir-MoO₃, IrO₂-MoO₃, and IrO₂. **b,** Derivative of L₃-edge XANES spectra for Ir foil, Ir-MoO₃, IrO₂-MoO₃, and IrO₂.

We newly added one sentence in our revised manuscript as below:

Note that Ir L₃-edge of IMO is also positively shifted compared with that of Ir metal foil due to the high electron-deficient surface of IMO (Figure 4a).

(1-4) For practical usage, a durability test under higher current density (e. g. >50 mA/cm²) is recommended to be provided for comparison in the work.

Answer 1-4) Thanks for the constructive comment. By following the review's comments, we newly conducted the durability test under a high current density of 100 mA/cm² for IMO. As expected, IMO showed good durability in the high current, which is promising for practical usage.

Figure S18. Chronopotentiometric curves of Ir-MoO₃ at a current density of 100 mA cm⁻².

We newly added Fig. S18 in the revised manuscript as below:

Further, the chronopotentiometry test at a high current density of 100 mA cm^{-2} for 48 h demonstrates the exceptional stability of IMO, which is promising for practical usage (Figure S18).

(1-5) In the part of choosing active sites for OER, the OH^* should be a more proper adsorption species for testing rather than O_2 , because the OH^* is the first intermediate formed in OER.

Answer 1-5) Thanks for the constructive comment. We totally agree with the review's comments.

We newly modified by choosing active sites using the HO^* rather than O_2 in revised Figures 6a and 6b as below.

Figure 6. a, The optimal sites for the HO^* adsorption on the interfacial site (A site) between the MoO_3 and Ir, Ir's edge site (B site), Ir's hollow site (C site), and Ir's top site (D site). **b**, The corresponding adsorption energies of the four sites in Figure 6a.

By following reviews' comments, firstly, we checked the favorite sites for HO^* adsorption at various IMO sites since the HO^* is the first intermediate formed in the OER (Figure 6a). The

interfacial site (A site) between the MoO₃ and Ir, the Ir edge site (B site), the Ir hollow site (C site), and the Ir top site (D site) were calculated for adsorption energy of HO*, demonstrating the sites A, B and C are favorable for the HO* adsorption compared to the D site (**Figure 6b**). As a result, we found the HO* is possible to adsorb the site A, B, and C. D is unstable during the optimized process. Based on the result, we newly revised the DFT part of this manuscript as below:

Firstly, we check the favorite sites for HO* adsorption at various IMO sites since the HO* is the first intermediate formed in the OER (**Figure 6a**). The interfacial site (A site) between the MoO₃ and Ir, the Ir edge site (B site), the Ir hollow site (C site), and the Ir top site (D site) are calculated for adsorption energy of HO*, demonstrating the sites A, B and C are favorable for the HO* adsorption compared with the D site (**Figure 6b**). Note that the D site on IMO is unstable during the optimized process, which prevents the D site from becoming the active site for the OER process. Additionally, note that the same trend of oxygen adsorption was observed and the D site on IMO showed the bond distance of 2.72 Å (**Figure S20**), indicating that the D site is not a favorable site for oxygen adsorption. Thus, we adopt sites A, B, and C for further study of site activity mechanisms.

(1-6) Some relevant works are recommended to be cited: Advanced Functional Materials, 2020, 1908708; Nature Communications, 2021, 12, 2351; Nature Energy, 2019, 4, 329; Advanced Functional Materials, 2018, 1805227

Answer 1-6) Thanks for the constructive suggestion. We totally agree with the reviewer's comment to cite the valuable literatures. By following the comments, **we cited those articles in three parts**, as shown below.

(1) Hydrogen (H₂) fuel, as a clean energy carrier, is promising to provide an environmentally benign solution for global energy needs (*Advanced Functional Materials*, 2020, 1908708; *Nature Communications*, 2021, 12, 2351).

(2) Xin Wang and coworkers proposed a lattice oxygen oxidation mechanism pathway using metal oxyhydroxides when two adjacent oxidized oxygen atoms can hybridize their oxygen holes without sacrificing metal-oxygen hybridization (*Nature Energy*, 2019, 4, 329).

(3) The annealing procedure at 500 degrees in the air can induce the electron-deficient surface of metallic Ir due to the surface oxygen and heterointerface junction with the semiconductor of MoO₃ (*Advanced Functional Materials*, 2018, 1805227).

Reviewer #2

Reviewer #2: In this manuscript X. Liu et al. fabricated and characterized an Ir based IMO electrocatalyst for the OER. They found that these catalysts present higher OER activity at low over-potential as well as higher activities under reaction conditions. The manuscript is well written, and the results are well supported. However, there are some issues that the authors should consider in order to improve the quality of this manuscript:

Response: We are grateful for the time and effort. Reviewer 2 has spent reviewing our manuscript.

The review comments are constructive and valuable to improve our manuscript.

(2-1) Can the author include some reference for a better comparison with the acquired spectra in this research (i.e. Ir₀ and IrO₂ (IV))? Ideally one should consider to collect both XPS and NEXAFS.

Answer 2-1) Yes, Thanks for the constructive comment. We totally agree with the review comments.

For the XAFS, we newly added the references of Ir foil and IrO₂ as shown below: We added the comparison of Ir L₃-edge between IMO and Ir metal, as shown in revised Figures 4a-c. As we expected, the Ir L₃-edge was also positively shifted compared with the Ir metal, which is consistent with the results of Mo K-edge in IMO.

Fig. 4 | X-ray absorption fine structure (XAFS) characterizations. a, XANES survey spectra at the Ir L₃-edge for Ir foil, Ir-MoO₃, IrO₂-MoO₃, and IrO₂. **b**, Derivative of L₃-edge XANES spectra for Ir foil, Ir-MoO₃, IrO₂-MoO₃, and IrO₂. **c**, Ir L₃-edge X-ray absorption Fourier-transformed (FT) k³-weighted EXAFS spectra in R space.

We newly added one sentence in revised our manuscript as “Note that Ir L₃-edge of IMO is also positively shifted compared with that of Ir metal foil due to high electron-deficient surface of IMO (Figure 4a)”

For the XPS, we newly added the references of Ir foil and IrO₂ as shown below:

Figure S10. HR-XPS for the Commercial Ir, Commercial IrO₂, and IMO.

Fig. 3 | Electron transfer behaviors characterizations of IMO. **a**, HR-XPS of Ir 4f.

As a result, the IMO still has a high energy shift in comparison with commercial IrO₂, which is consistent with the results of IOMO. We newly added one sentence in our revised manuscript as “We confirm that the Ir NPs of the IMO has an electron-deficient surface, as evidenced by a higher Ir surface valence state of IMO than that of IOMO and commercial IrO₂ (Figures 3a and S10).”

(2-2) For this reviewer is difficult to follow what are Figures 2g and 2f. These images are better explained in Fig. S5. The authors should consider to swap images from the SI to the main text and remove Fig S5 from the SI.

Answer 2-2) Yes, we thank the reviewer for raising the issue. We totally agree with the review's comments. By following reviews comments, **we swapped images from the Fig. S5 of SI to Fig.2 of the main text and removed Fig S5 from the SI.** Thanks a lot for your constructive comments. We newly revised in the revised manuscript.

Fig. 2 | Morphology and structural characterizations of IMO. f, and g, HRTEM image.

(2-3) The EELS spectra are of bad quality. The authors should consider to plot another EELS spectra with better quality or remove these spectra from the text. The information provided by the X-ray spectroscopy should be enough.

Answer 2-3) Yes, we thank the reviewer for raising the issue. We totally agree with the review's comments. By following the review's comments, we removed the **EELS spectra from the text because the M-edge of Ir has high energy, which is almost beyond the energy range (300 – 2,000 eV) of EELS** (Ref: *EELS in the STEM: Determination of materials properties on the atomic scale*; [https://doi.org/10.1016/S0968-4328\(97\)00033-4](https://doi.org/10.1016/S0968-4328(97)00033-4)). Thus, it is a challenge to obtain the high-quality signal of Ir in this current stage. So, by following the review's comments, we removed the EELS spectra in the revised manuscript. We strongly believe that the XPS and XAFS spectra are enough to support our statements. Thanks again for your constructive comments.

(2-4) The XPS spectra fitting is not optimal. It is necessary to re-fit the whole set of spectra, indicate the parameters used (peak position, peak shape and fwhm), and justify why these peaks were used to fit the spectra. In this point the reference spectra can help significantly. In addition it will be nice to show the C1s and VB. Furthermore, it should be indicated how the spectra were calibrated.

Answer 2-4) Yes, we thank the reviewer for raising the issue.

Firstly, we **re-fitted the whole set of spectra of XPS**. To clearly demonstrate the peak position, peak shape, and FWHM, we newly made the table S1, as shown below. To justify these peaks that we used for fitting, we newly added the reference spectra of commercial Ir metal and IrO₂. Additionally, the intensity ratio was fixed using standard ratios of $3d_{3/2} : 3d_{5/2} = 2 : 3$ and $4f_{5/2} : 4f_{7/2} = 3 : 4$, which can be helpful to justify the peaks used to fit in those spectra.

Table S1. Parameter for deconvolution of XPS spectra.

Material	Peak	Position (eV)	Shape (eV)	FWHM* (eV)	Peak Area
IMO	Ir 4f _{7/2}	61.93	Gaussian	1.4	2239.5
	Ir 4f _{5/2}	64.84	Gaussian	1.3	1710.3
	O 1s	530.31	Gaussian	1.4	11418.9
	O 1s	531.36	Gaussian	1.4	5334.9
	O 1s	532.68	Gaussian	1.6	1379.1
	Mo ⁵⁺ 3d _{5/2}	231.72	Gaussian	1.6	2860.9
	Mo ⁶⁺ 3d _{5/2}	232.79	Gaussian	1.46	12835.1
	Mo ⁵⁺ 3d _{3/2}	234.46	Gaussian	1.2	1514.3
	Mo ⁶⁺ 3d _{3/2}	235.89	Gaussian	1.4	8539.4
IOMO	Ir 4f _{7/2}	61.55	Gaussian	1.3	1181.2
	Ir 4f _{5/2}	64.53	Gaussian	1.22	917.7
	O 1s	530.33	Gaussian	1.5	10722.1
	O 1s	531.36	Gaussian	1.6	3366.9
	O 1s	532.66	Gaussian	1.4	739.1
	Mo ⁵⁺ 3d _{5/2}	232.10	Gaussian	1.2	805.8
	Mo ⁶⁺ 3d _{5/2}	232.59	Gaussian	1.3	9068.9
	Mo ⁵⁺ 3d _{3/2}	234.18	Gaussian	1.04	543.97
Mo ⁶⁺ 3d _{3/2}	235.70	Gaussian	1.3	6078.51	
Ir	Ir 4f _{7/2}	60.93	Gaussian	1.2	3360.0
	Ir 4f _{5/2}	63.83	Gaussian	1.23	2583.7
IrCl ₃	Ir 4f _{7/2}	61.68	Gaussian	1.8	967.8
	Ir 4f _{5/2}	64.67	Gaussian	1.7	727.1
IrO ₂	Ir 4f _{7/2}	61.63	Gaussian	1.34	1607.1
	Ir 4f _{5/2}	64.56	Gaussian	1.4	1204.2
	O 1s	529.83	Gaussian	1.4	1385.1
	O 1s	531.18	Gaussian	1.8	1117.2
	O 1s	533.05	Gaussian	1.8	464.95

*FWHM: Full with at half maximum.

For energy calibration, all XPS spectra were calibrated using C 1s at 284.6 eV, which also was added in Fig. S7 in the experimental section of the revised manuscript. In fact, all spectra of XPS were measured in the Cooperative Center for Research Facilities (CCRF) of SKKU and the expert operator always has used C 1s as the standard process for spectra calibrations, which is a general process for the technology of XPS calibration. We strongly believe that the C 1s is valid

for the energy calibration, demonstrating the C 1s position for all spectra in Figure S7, which indicated that the energy of spectra was properly corrected.

Figure S7. The survey spectrum with energy calibration using C 1s at 284.6 eV. a, Commercial Ir. b, Commercial IrO₂. c, IrO₂-MoO₃. d, Ir-MoO₃.

(2-5) Figure 5e should be plotted without the XPS spectra for clarity.

Answer 5) Yes, Thanks for the constructive comment. We agree with the review's comments and we deleted the XPS spectra in Fig. 5 in revised manuscript and also the deleted XPS spectra of Fig. 5 was moved as Figure S19 in the revised Supporting information.

Fig. 5 | Electrochemical oxygen evolution reaction (OER) performance. e, Chronopotentiometric curves of Ir-MoO₃, RuO₂, and Ir.

Figure S19. XPS patterns for before and after the OER stability measurement. a, IMO and b, IOMO.

(2-6) I cannot see Figure S14 in the SI.

Answer 2-6) Yes, we thank the reviewer for raising the issue. We are so sorry for our mistake. When the pdf file was converted from the word file, Fig. S14 was not generated. **We carefully checked all figures properly at this time** when we converted the word file to a pdf file. In this new version of our manuscript, Figure S14 was re-named as Figure S19, as shown below:

Figure S19. XPS patterns for before and after the OER stability measurement. a, IMO and b, IOMO.

Reviewer #3: The paper of Liu et al. presents synthesis and characterization of metal/MoO₃ catalysts. The materials were prepared using electrospinning synthetic method and analyzed by microscopic and spectroscopic methods, while their oxygen evolution activity and durability were studied in an RDE set-up. The DFT calculations on different IR sites were also provided in this work.

However, this paper has to be reviewed before publishing, specifically the following points has to be addressed:

Response: We are grateful for the time and effort. Reviewer 3 has spent reviewing our manuscript

and agree published after addressing the comments. The review comments are constructive and valuable to improve our manuscript.

(3-1-1) energy calibration in XPS and XANES analysis was not mentioned in the experimental section, while it is very important while discussing the spectra shifts. (3-1-2) Are the observed shifts in XPS related to charging (MoO₃ as a semiconductor may lead to this)? (3-1-3) In case of XANES, the Ir edge (metal) is at 11.21520 keV, that does not correspond to the metallic Ir provided in the manuscript

Answer 3-1-1) We thank you for the constructive comment. We totally agree with the review comments. We newly added the energy calibration in XPS and XANES analysis in the experimental section, as shown below:

For energy calibration of XPS, **all XPS spectra were calibrated using C 1s at 284.6 eV**, which also was added in the experimental section of the revised manuscript. In fact, all spectra of XPS were measured in the Cooperative Center for Research Facilities (CCRF) of SKKU and the expert operator always uses C 1s as the standard process for spectra calibrations, which is a general process for the technology of XPS calibration. We strongly believe that the C 1s is valid for the energy calibration, demonstrating the C 1s position for all spectra in Figure S7, which indicates that the energy of spectra was properly corrected.

Figure S7. The survey spectrum with energy calibration using C 1s at 284.6 eV. a, Commercial Ir. b, Commercial IrO₂. c, IrO₂-MoO₃. d, Ir-MoO₃.

We newly added the detailed information as “X-ray photoelectron spectroscopic (XPS) measurements were performed on a VG Microtech ESCA 2000 using a monochromic Al X-ray source (97.9 W, 93.9 eV); Note that all XPS spectra were calibrated using C 1s at 284.6 eV.”

For energy calibration of XANES, the energy was calibrated by setting the first inflection points of reference metal foils to the absorption edge energy as 11.215 keV for Ir-L3 edge and 20 keV for Mo K-edge.

We newly added the detailed information as “Ir L₃-edge and Mo K-edge XAFS experiments were conducted in an ambient condition at 10C beamline of Pohang Light Sources-II (PLS-II). The incident beam was monochromatized using a Si (111) double crystal monochromator and detuned to ~70 % of its maximum intensity for reducing higher-order harmonics. For energy calibration, the first inflection points of reference metal foils were set to the corresponding absorption edge energy (11.2152 keV for Ir-L3 edge and 19.995 keV for Mo K-edge). After normalization, the EXAFS signal was k³-weighted to magnify high-energy oscillations and Fourier-transformed in the k-range from 2.5 to 11.5 Å⁻¹.”

For the question of (3-1-2), Are the observed shifts in XPS related to charging (MoO₃ as a semiconductor may lead to this)?

Answer 3-1-2) Yes, we agree with the reviews' comments. As we mentioned in our original manuscript, we observed Ir 4f shifts in XPS for (i) surface oxygen of Ir and (ii) the electron-withdrawing material of MoO₃. For the second factor, the charges (electrons) of Ir NPs are transferred to MoO₃, evidenced by the charge density difference simulation (**Figures 3c and 3d**) and the Mo⁵⁺ from the XPS (**Figure 3b**).

Fig. 3 | Electron transfer behaviors characterizations of IMO. a, HR-XPS of Ir 4f. b, HR-XPS of Mo 3d. c, the charge density difference of IMO. d, The enlarged charge density difference for Figure 3c. The yellow and cyan regions represent electron accumulation and depletion, respectively. The red, gold, and violet balls represent the O, Ir, and Mo atoms, respectively. The isosurface value is 0.015e/bohr³.

For the question of (3-1-3), In case of XANES, the Ir edge (metal) is at 11.21520 keV, that does not correspond to the metallic Ir provided in the manuscript.

Answer 3-1-3) Thank the reviewer for raising the issue. We are sorry for our mistakes when we plotted the Figures. We corrected the plot that is corresponded to 11.2152 keV for Ir-L3 edge and revised all the Figures (Figs. 4a and 4b) based on the energy calibration mentioned above, as shown below:

Fig. 4 | X-ray absorption fine structure (XAFS) characterizations. a, XANES survey spectra at the Ir L₃-edge for Ir foil, Ir-MoO₃, IrO₂-MoO₃, and IrO₂. b, Derivative of L₃-edge XANES spectra for Ir foil, Ir-MoO₃, IrO₂-MoO₃, and IrO₂.

3-2. Analysis of XPS spectra. The comparison of Ir and Mo core-levels cannot be done when the fitting parameters are not consistent (the intensity ratio are not fixed, the relative binding energy etc). Please correct this.

Answer 3-2) We thank you for the constructive comment. **We re-fitted the whole set of spectra of XPS.** To clearly demonstrate the peak position, peak shape, and FWHM, we newly made the table S1, as shown below. To justify these peaks that we used for fitting, we added the reference spectra of commercial Ir metal and IrO₂. Additionally, the intensity ratio was fixed using standard ratios of 3d_{3/2}: 3d_{5/2} = 2 : 3 and 4f_{5/2} : 4f_{7/2} = 3 : 4, which can be helpful to justify the peaks used to fit in those spectra.

Table S1. Parameter for deconvolution of XPS spectra.

Material	Peak	Position (eV)	Shape (eV)	FWHM* (eV)	Peak Area/%
IMO	Ir 4f _{7/2}	61.93	Gaussian	1.4	2239.5
	Ir 4f _{5/2}	64.84	Gaussian	1.3	1710.3
	O 1s	530.31	Gaussian	1.4	11418.9
	O 1s	531.36	Gaussian	1.4	5334.9
	O 1s	532.68	Gaussian	1.6	1379.1
	Mo ⁵⁺ 3d _{5/2}	231.72	Gaussian	1.6	2860.9
	Mo ⁶⁺ 3d _{5/2}	232.79	Gaussian	1.46	12835.1
	Mo ⁵⁺ 3d _{3/2}	234.46	Gaussian	1.2	1514.3
	Mo ⁶⁺ 3d _{3/2}	235.89	Gaussian	1.4	8539.4
IOMO	Ir 4f _{7/2}	61.55	Gaussian	1.3	1181.2
	Ir 4f _{5/2}	64.53	Gaussian	1.22	917.7
	O 1s	530.33	Gaussian	1.5	10722.1
	O 1s	531.36	Gaussian	1.6	3366.9
	O 1s	532.66	Gaussian	1.4	739.1
	Mo ⁵⁺ 3d _{5/2}	232.10	Gaussian	1.2	805.8
	Mo ⁶⁺ 3d _{5/2}	232.59	Gaussian	1.3	9068.9
	Mo ⁵⁺ 3d _{3/2}	234.18	Gaussian	1.04	543.97
	Mo ⁶⁺ 3d _{3/2}	235.70	Gaussian	1.3	6078.51
Ir	Ir 4f _{7/2}	60.93	Gaussian	1.2	3360.0
	Ir 4f _{5/2}	63.83	Gaussian	1.23	2583.7
IrCl ₃	Ir 4f _{7/2}	61.68	Gaussian	1.8	967.8
	Ir 4f _{5/2}	64.67	Gaussian	1.7	727.1
IrO ₂	Ir 4f _{7/2}	61.63	Gaussian	1.34	1607.1
	Ir 4f _{5/2}	64.56	Gaussian	1.4	1204.2
	O 1s	529.83	Gaussian	1.4	1385.1
	O 1s	531.18	Gaussian	1.8	1117.2
	O 1s	533.05	Gaussian	1.8	464.95

*FWHM: Full width at half maximum.

3-3. The authors claim the formation of metallic Ir in case of IMO sample where there is no spectroscopic evidence, it looks like it is an oxide, however the oxidation state is lower than IV. Moreover, the analysis of Ir4f XPS spectra shows higher binding energy in comparison to IrO₂, that is related to Ir (III) oxidation state as it was discussed in many publications previously (exception to a common situation higher BE = higher oxidation state).

Answer 3-3) We thank the reviewer for giving constructive comments. We agree with the review's comments. We newly added Table S1 in the revised Supporting Information. As the review mentioned, **our measurement of newly added Table S1 showed that Ir (III) oxidation state is higher than the IrO₂**, as below:

Material	Peak	Position (eV)	Shape (eV)	FWHM* (eV)	Peak Area/%
IMO	Ir 4f _{7/2}	61.93	Gaussian	1.4	2239.5
	Ir 4f _{5/2}	64.84	Gaussian	1.3	1710.3
	O 1s	530.31	Gaussian	1.4	11418.9
	O 1s	531.36	Gaussian	1.4	5334.9
	O 1s	532.68	Gaussian	1.6	1379.1
	Mo ⁵⁺ 3d _{5/2}	231.72	Gaussian	1.6	2860.9
	Mo ⁶⁺ 3d _{5/2}	232.79	Gaussian	1.46	12835.1
	Mo ⁵⁺ 3d _{3/2}	234.46	Gaussian	1.2	1514.3
	Mo ⁶⁺ 3d _{3/2}	235.89	Gaussian	1.4	8539.4
IrCl ₃	Ir 4f _{7/2}	61.68	Gaussian	1.8	967.8
	Ir 4f _{5/2}	64.67	Gaussian	1.7	727.1
IrO ₂	Ir 4f _{7/2}	61.63	Gaussian	1.34	1607.1
	Ir 4f _{5/2}	64.56	Gaussian	1.4	1204.2
	O 1s	529.83	Gaussian	1.4	1385.1
	O 1s	531.18	Gaussian	1.8	1117.2
	O 1s	533.05	Gaussian	1.8	464.95

However, we are sure that our Ir metals in IMO show the high surface valence state, which is not metal oxide, with the evidence as below:

Reason 1: The phase structures of IMO show that the phase properties of iridium are metallic, evidenced by the XRD pattern. Because the Ir NPs in IMO are matched with Ir (original **Figure 2a**), but not matched with Ir(III) (**Figure R4**).

Figures 2a. The PXRD pattern of IMO.

Figures R4. The XRD pattern of IrCl₃.

Reason 2: The phase structures of IMO show that the phase properties of iridium are metallic because the Ir properties are almost similar to Ir foil, evidenced from the EXAFS spectra and WT-EXAFS images, as shown below. We added EXAFS spectra of Ir-foil in Fig. 4c and WT-EXAFS images in Fig. 4g.

Figure 4c, Ir L3-edge X-ray absorption Fourier-transformed (FT) k^3 -weighted EXAFS spectra in R space.

Figure 4g. Wavelet transforms (WT-EXAFS) for Ir-MoO₃ and Ir.

Reason 3: To uncover which part of Ir NPs in IMO shows the valance state higher than 4^+ , the high-resolution XPS measurement was conducted before and after Ar etching experiment (*Energy Environ. Sci.*, 2020, 13, 5152-5164). As we expected, the Ir metallic property of Ir NPs in IMO was revealed after argon etching (**Figure S11**), which is consistent with the commercial Ir metal. The oxygen attached to the Ir surface was removed by the Ar etching and finally gave Ir metallic property, which is consistent with XRD and EXAFS spectra. We newly added Figure S11.

Figure S11. HR-XPS of Ir 4f for the Commercial Ir, IMO, and argon (Ar) etching IMO.

In addition, we newly added a sentence in our revised manuscript as “To prove whether the interfacial Ir atoms binding to MoO₃ or the surface Ir atoms binding to adsorbed oxygen species have higher valence states, the HR-XPS measurement about Ir 4f in IMO is performed with an argon ion etching treated sample⁴¹, demonstrating zero-valence state of Ir metal in IMO, which is consistent with commercial Ir metal (Figure S11) and XRD pattern (Figure 2a). This result confirms that the surface Ir in IMO has a high valence state.”

Based on the three reasons above, we believe that Ir in the IMO sample has a high valence state due to the high electron-deficient state of Ir metal, which is not directly related to the IrCl₃.

3-4. The analysis of the electrochemical data shows higher activity of the synthesized materials, specifically for IMO. The Tafel slope of 48 mV dec⁻¹ in case of IMO in comparison to other materials with 70-80 mV dec⁻¹ should correspond to a change in the rds, however nothing is discussed in the manuscript.

Answer 3-4) We thank the reviewer for giving constructive comments. By following the review's comments, we newly added more explanations in the part of Tafel slope, as shown below:

By following the previous literature (*Sci Rep* 5, 13801 (2015).), according to Tafel slope value, the Tafel slope of 120 mV dec⁻¹ was observed when the surface species formed in the step just before the rate-determining step was predominant. In the other cases, the Tafel slope was lower than 120 mV dec⁻¹. When the surface adsorbed species produced in the early stage of the OER remained predominant, the Tafel slope decreased. In particular, a low theoretical Tafel slope of ~ 30 mV dec⁻¹ could be observed with high coverage (> 60%) of the empty sites by the active species (such as O and OH species), which was experimental valid from some references (Fe₅₀Co₅₀O_x,

Fe₅₀Ni₅₀O_x and Fe₃₃Co₃₃Ni₃₃O_x with approximately Tafel slope of 30 mV dec⁻¹, *ref: Science 340, 60 (2013)* and NiFe-LDH with Tafel slope of 35 mV dec⁻¹, *ref: J. Am. Chem. Soc. 135, 8452 (2013)*).

The IMO showed the low Tafel slope of 48 mV dec⁻¹ due to a large number of oxygen species. **The absorbed oxygen species not only gave the high valence state surface of Ir in IMO but also decreased the Tafel slope, which could accelerate the process of OER and increase the OER efficiency.** In addition, based on the DFT simulation, when the oxygen species adopted the surface of Ir metal, the proton dissociating pathway was more preferred because it could further decrease the energy barrier and break the scaling relationship between HOO* and HO*.

Thus, by following the review's constructive suggestion, we newly added the discussion in our revised manuscript as “**The Tafel slope of IMO is 48 mV dec⁻¹, which is remarkably lower than the state-of-the-art RuO₂ at 75 mV dec⁻¹, consistent with the previous reports^{1,2} and Ir at 71 mV dec⁻¹, consistent with the previous reports². All measured values of Tafel slope is lower than 120 mV dec⁻¹ and we can conclude surface species formed in the step just before the rate-determining step is not predominant.³ Due to the high coverage of active species at empty sites that decrease the value of Tafel slope^{3,4,5}, the low Tafel slope of IMO can be attributed to a large number of oxygen species on surface Ir in IMO. The absorbed oxygen species not only provide the high valence state surface of Ir in IMO but also reduce the value of Tafel slope, which accelerates the process of OER and increases the OER efficiency.**”

“In addition, to break the scaling relation between HOO* and HO*^{6,7}, the proton dissociation pathway (PDP) is suggested for the models IMO (O-8). As a result, PDP indicates the lowest

energy barrier when the proton is transferred to the neighbor surface oxygen compared to the relative other OER pathway, and the corresponding configurations are illustrated (Figure 6c), which is consistent with the experimental results of low overpotential and Tafel slope. These results significantly demonstrate that the surface oxygen participates in the OER reaction as a proton acceptor, powerfully uncovering the origin of IMO's excellent catalytic OER performance, which opens up a new avenue for designing highly efficient catalysts.”

3-5. The durability analysis of IOMO (Figure 5e) shows an interesting change at 20h, what is happening to the sample ? Was it analyzed ? This should be discussed in the manuscript and compared with IMO behavior.

Answer 3-5) We thank the reviewer giving the constructive comments. By following the reviews guidance, we measured the XPS after the stability test for the samples of IMO and IOMO, as shown in new Figure S19. The fitting parameters were newly shown in Table S4.

Based on the XPS result, we can clearly see that the surface structure of IOMO was corroded since the lattice oxygen ratio sharply decreased. Even the Ir 4f of used IOMO ($4f_{7/2}$: 62.51 eV; $4f_{5/2}$: 65.38 eV) showed the low valence state compared with that of used IMO ($4f_{7/2}$: 62.64 eV; $4f_{5/2}$: 65.64 eV), the surface structure of IOMO was destroyed since the its lattice oxygen sharply decreased after OER test (Table S4). This result showed the IrO_2 in IOMO was unstable in comparison with the high valence state of Ir in IMO, which is the origin of the high activity and stability for IMO.

Figure S19. XPS patterns for before and after the OER stability measurement. **a**, IMO and **b**, IOMO.

We newly added this part in the revised manuscript as follows. “To gain insight into the mechanism, we measure the XPS spectra after the OER experiment. The energy state of the Ir, Mo, and O underwent a minor high-energy shift after the OER process, indicating that these elements lost the electron during the OER (Figure S19 and Table S4). Note that the surface structure of the IMO is still preserved well after the OER reaction since the synergic effect of the high surface state of Ir with the help of the Mo^{5+} can withstand resistance in an oxidation state (Figure S19a). However, the surface structure of IOMO can be corroded during the OER process because the ratio of lattice oxygen (O_L) decreases compared with that in fresh of IOMO, (Figure S19a), which is the origin of the instability for IOMO in comparison with IMO.”

Table S4. Parameter for deconvolution of XPS spectra before/after the OER reaction for IMO and IOMO.

Material	Peak	Position (eV)	Shape (eV)	FWHM* (eV)	Peak Area
IMO	Ir 4f _{7/2}	61.93	Gaussian	1.4	2239.5
	Ir 4f _{5/2}	64.84	Gaussian	1.3	1710.3
	O 1s	530.31	Gaussian	1.4	11418.9
	O 1s	531.36	Gaussian	1.4	5334.9
	O 1s	532.68	Gaussian	1.6	1379.1
	Mo ⁵⁺ 3d _{5/2}	231.72	Gaussian	1.6	2860.9
	Mo ⁶⁺ 3d _{5/2}	232.79	Gaussian	1.46	12835.1
	Mo ⁵⁺ 3d _{3/2}	234.46	Gaussian	1.2	1514.3
	Mo ⁶⁺ 3d _{3/2}	235.89	Gaussian	1.4	8539.4
Used IMO	Ir 4f _{7/2}	62.64	Gaussian	1.7	358.15
	Ir 4f _{5/2}	65.64	Gaussian	1.74	268.82
	O 1s	530.54	Gaussian	1.5	11838.4
	O 1s	531.36	Gaussian	1.4	2753.67
	O 1s	532.46	Gaussian	1.45	1139.73
	Mo ⁵⁺ 3d _{5/2}	231.94	Gaussian	1.5	741.41
	Mo ⁶⁺ 3d _{5/2}	233.24	Gaussian	1.5	1879.54
	Mo ⁵⁺ 3d _{3/2}	234.60	Gaussian	1.4	498.31
	Mo ⁶⁺ 3d _{3/2}	236.23	Gaussian	1.8	1262.28
IOMO	Ir 4f _{7/2}	61.55	Gaussian	1.3	1181.2
	Ir 4f _{5/2}	64.53	Gaussian	1.22	917.7
	O 1s	530.33	Gaussian	1.5	10722.1
	O 1s	531.36	Gaussian	1.6	3366.9
	O 1s	532.66	Gaussian	1.4	739.1
	Mo ⁵⁺ 3d _{5/2}	232.10	Gaussian	1.2	805.8
	Mo ⁶⁺ 3d _{5/2}	232.59	Gaussian	1.3	9068.9
	Mo ⁵⁺ 3d _{3/2}	234.18	Gaussian	1.04	543.97
	Mo ⁶⁺ 3d _{3/2}	235.70	Gaussian	1.3	6078.51
Used IOMO	Ir 4f _{7/2}	62.51	Gaussian	1.6	1499.94
	Ir 4f _{5/2}	65.38	Gaussian	1.4	1127.94
	O 1s	530.57	Gaussian	1.4	1829.78
	O 1s	531.70	Gaussian	1.6	4796.47
	O 1s	532.83	Gaussian	1.4	6277
	Mo ⁵⁺ 3d _{5/2}	232.54	Gaussian	1.4	424.23
	Mo ⁶⁺ 3d _{5/2}	233.27	Gaussian	1.24	2217.66
	Mo ⁵⁺ 3d _{3/2}	234.45	Gaussian	1.2	280.78
	Mo ⁶⁺ 3d _{3/2}	236.41	Gaussian	1.34	1476.57

3-6. Based on the DFT data, the authors proposed the OER mechanism on the synthesized materials, however it is not well discussed and not clear to a reader - has to be improved.

Answer 3-6) We thank the reviewer for the constructive comments. For more well discussed and clear to the reader, we added the HO* adsorption part and revised the OER mechanism part as below.

“Firstly, we check the favorite sites for HO* adsorption at various IMO sites since the HO* is the first intermediate formed in the OER (Figure 6a). The interfacial site (A site) between the MoO₃ and Ir, the Ir edge site (B site), the Ir hollow site (C site), and the Ir top site (D site) are calculated as favorable for adsorption energy of HO*, demonstrating the sites A, B and C are favorable for the HO* adsorption compared to the D site (Figure 6b). Note that the D site on IMO is unstable during the optimized process, which prevents the D site from becoming the active site during the OER process. Additionally, note that the same trend of oxygen adsorption is observed and the D site on IMO shows the bond distance of 2.72 Å (Figure S20), indicating that the D site is not a favorable site for oxygen adsorption. Thus, we adopt sites A, B, and C to further study the mechanism of site activity. With its electron-deficient surface, IMO has a high OER catalytic efficiency and fast kinetic processes, resulting in the boosted OER performance^{1, 8}. The B site is more favored for the OER process compared to sites A and C because the configuration of HOO* cleaves as O* and HO* during the optimization process at sites of A and C sites (Figures 6d-f). Although the intermediate configuration of HOO* at site B is stable (Figures 6c-d and Figure S21), but the energy barrier is still high.

To gain insight into the OER mechanism of IMO by considering its boosting OER efficiency, the various amount of surface oxygen species was considered (Figures S22-26). As a result, the Ir

metallic surface with 8 surface oxygen atoms has the highest formation energy (-1.74 eV/atom) compared to other amount surface oxygen atoms (-1.63 eV/atom for 7 surface oxygen; -1.62 eV/atom for 9 surface oxygen and -1.42 eV/atom for 10 surface oxygen). As we expected, the IMO with 7 surface oxygen atoms (O-7) models shows the low OER energy barriers by reducing the energy barriers compared with the IMO (Figures 6c and S27). In addition, to break the scaling relation between HOO^* and $\text{HO}^{*14, 55}$, the proton dissociation pathway (PDP) is suggested for the models IMO (O-8). As a result, PDP indicates the lowest energy barrier when the proton transfers to the neighbor surface oxygen compared to the relative other OER pathway, and the corresponding configurations are illustrated (Figure 6c), which is consistent with the experimental results of low overpotential and Tafel slope. These results significantly demonstrate that the surface oxygen participates in the OER reaction as a proton acceptor, powerfully uncovering the origin of IMO's excellent catalytic OER performance, which opens up a new avenue for designing highly efficient catalysts.”

Fig. 6 | DFT computational modeling and investigation of IMO active site. **a**, The optimal sites for the HO* adsorption on the interfacial site (A site) between the MoO₃ and Ir, Ir's edge site (B site), Ir's hollow site (C site), and Ir's top site (D site). **b**, The corresponding adsorption energies of the four sites in Figure 6a. **c**, Relative energy profiles, and the simplified surface structures of various reaction species as the arrow define. **d**, The configuration of HOO* for site B after relaxation. **e**, The configuration of HOO* for site A after relaxation. **f**, The configuration of HOO* for site C after relaxation. Color code: The red, gold, and violet balls represent the O, Ir, and Mo atoms, respectively.

Reference

1. Lee J, *et al.* Stabilizing the OOH* intermediate via pre-adsorbed surface oxygen of a single Ru atom-bimetallic alloy for ultralow overpotential oxygen generation. *Energy & Environmental Science* **13**, 5152-5164 (2020).
2. Tran NQ, *et al.* Low Iridium Content Confined inside a Co₃O₄ Hollow Sphere for Superior Acidic Water Oxidation. *Acs Sustainable Chemistry & Engineering* **7**, 16640-16650 (2019).
3. Shinagawa T, Garcia-Esparza AT, Takanabe K. Insight on Tafel slopes from a microkinetic analysis of aqueous electrocatalysis for energy conversion. *Scientific Reports* **5**, 13801 (2015).
4. Gong M, *et al.* An Advanced Ni–Fe Layered Double Hydroxide Electrocatalyst for Water Oxidation. *JACS* **135**, 8452-8455 (2013).
5. Smith RDL, *et al.* Photochemical Route for Accessing Amorphous Metal Oxide Materials for Water Oxidation Catalysis. *Science* **340**, 60 (2013).
6. Song J, *et al.* A review on fundamentals for designing oxygen evolution electrocatalysts. *Chemical Society Reviews* **49**, 2196-2214 (2020).
7. Pham HH, Cheng M-J, Frei H, Wang L-W. Surface Proton Hopping and Fast-Kinetics Pathway of Water Oxidation on Co₃O₄ (001) Surface. *ACS Catalysis* **6**, 5610-5617 (2016).
8. Velasco-Vélez JJ, *et al.* Electrochemically active Ir NPs on graphene for OER in acidic aqueous electrolyte investigated by in situ and ex situ spectroscopies. *Surface Science* **681**, 1-8 (2019).

REVIEWERS' COMMENTS

Reviewer #1 (Remarks to the Author):

The authors have addressed my concerns well and I can now recommend it to publish as is.

Reviewer #2 (Remarks to the Author):

Thanks for the resubmitted manuscript and for addressed the issues pointed in my previous report. There is only a minor issue that should be considered to be added in the re-submitted manuscript and it is related to the formation of Ir(III) during OER, as was pointed in the revised manuscript. I recommend the authors to cite the next work and use it to argue that the oxidation state of Ir during OER is IV or higher >IV but not III:

J. Am. Chem. Soc. 2021, 143, 32, 12524–12534

In this reference it is probed clearly that the oxidation state of Ir during OER is not Ir(III) because there is an increase in the intensity of the Ir-L3 white-line peak due to the formation of more electron-holes in the Ir 5d orbitals. This fact is accompanied by the formation of a new peak in the Ir 4f (XPS) at higher binding energies. Thus, the increase in the number of 5d electron-holes during the OER cannot support the existence of Ir(III) during OER. It is clear that the Ir L3-edge doesn't support the formation of Ir(III) during the OER, neither in this manuscript or in most of the literature available. There are other factors that can be argued against the existence of the Ir(III) during the OER using the O K-edge (not investigated in this manuscript) as the fact that Ir(III) should not yield any feature in the O K-edge in the t_{2g} orbitals because they are occupied and thus a transition to these orbitals is not possible in this oxidation state.

Finally, I would like to congratulate the authors for this nice piece of work.

Reviewer #3 (Remarks to the Author):

The authors provided responses to the comments and improved the manuscript accordingly. The manuscript can be published in Nature Communication.

Responses for the Nature Communications.

Manuscript ID: NCOMMS-21-23624-A

Title: Restructuring highly electron-deficient metal-metal oxides for boosting stability in acidic oxygen evolution reaction

We are grateful to the editor, editorial staff, and reviewers for their critical comments and valuable suggestions. The manuscript has been strictly modified and improved after addressing all the suggestions as listed below:

(The explanations to the comments from reviewers are shown in **blue color with yellow highlight**).

Reviewer(s)' Points-by-points responses to Author:

Reviewer #1

Reviewer #1: The authors have addressed my concerns well and I can now recommend it to publish as is.

Response: We are grateful for the time and effort of Reviewer 1 to improve our manuscript.

Reviewer #2

Reviewer #2: Thanks for the resubmitted manuscript and for addressed the issues pointed in my previous report. There is only a minor issue that should be considered to be added in the re-submitted manuscript and it is related to the formation of Ir(III) during OER, as was pointed in the revised manuscript. I recommend the authors to cite the next work and use it to argue that the oxidation state of Ir during OER is IV or higher >IV but not III:

J. Am. Chem. Soc. 2021, 143, 32, 12524–12534

In this reference it is probed clearly that the oxidation state of Ir during OER is not Ir(III) because there is an increase in the intensity of the Ir-L3 white-line peak due to the formation of more electron-holes in the Ir 5d orbitals. This fact is accompanied by the formation of a new peak in the Ir 4f (XPS) at higher binding energies. Thus, the increase in the number of 5d electron-holes during the OER cannot support the existence of Ir(III) during OER. It is clear that the Ir L3-edge doesn't support the formation of Ir(III) during the OER, neither in this manuscript or in most of the literature available.

There are other factors that can be argued against the existence of the Ir(III) during the OER using the O K-edge (not investigated in this manuscript) as the fact that Ir(III) should not yield any feature in the O K-edge in the t_{2g} orbitals because they are occupied and thus a transition to these orbitals is not possible in this oxidation state.

Finally, I would like to congratulate the authors for this nice piece of work.

Response: We are grateful for the time and effort of Reviewer 2. The review comments are very constructive and valuable to improve our manuscript. Thanks again for your time and effort to improve our work.

We agree with the reviewer's comment to cite the valuable literature. By following the comments, we cited this article in the part of the introduction, as shown below.

Recently, Juan-Jesús Velasco-Vélez et al. probed clearly that the oxidation state of Ir during OER is $>IV$ rather than that of Ir (III) because there is an increase in the intensity of the Ir- L_3 white-line peak due to the formation of more electron-holes in the Ir $5d$ orbitals.³² (Ref. No. 32, *J. Am. Chem. Soc.* 2021, 143, 32, 12524–12534)

Reviewer #3: The authors provided responses to the comments and improved the manuscript accordingly. The manuscript can be published in Nature Communication.

Response: We are grateful for the time and effort of Reviewer 3 to improve our manuscript.